# DNA methylation dynamics and dysregulation delineated by high-throughput profiling in the mouse

## Graphical abstract

## Authors

Wanding Zhou, Toshinori Hinoue,
Bret Barnes, ..., Brian H. Chen, Hui Shen,
Peter W. Laird

## Correspondence

wanding.zhou@pennmedicine.upenn.
edu (W.Z.),
hui.shen@vai.org (H.S.),
peter.laird@vai.org (P.W.L.)

## In brief

Infinium arrays are a cost-effective high-throughput DNA methylation profiling tool for human samples. Zhou et al. have developed an equivalent array for the mouse and used it to generate a DNA methylome atlas representing the diversity of DNA methylation biology in development, aging, and disease in this important model organism.

## Highlights

- Design, annotation, and validation of the Infinium Mouse DNA Methylation BeadChip

- Mouse methylome atlas of 1,239 diverse cell types, strains, ages, and pathologies

- Bioinformatics for tissues, imprinting, comparative epigenomics, strain SNPs, and PDXs

- Tumor- and age-associated DNA methylation dynamics and an epigenetic clock

 Zhou et al., 2022, Cell Genomics 2, 100144
July 13, 2022 © 2022 The Author(s).

# Cell Genomics

CellPress

## Resource

# DNA methylation dynamics and dysregulation delineated by high-throughput profiling in the mouse

Wanding Zhou,[1,2,*] Toshinori Hinoue,[3] Bret Barnes,[4] Owen Mitchell,[3] Waleed Iqbal,[1] Sol Moe Lee,[1] Kelly K. Foy,[3] Kwang-Ho Lee,[3] Ethan J. Moyer,[1] Alexandra VanderArk,[5] Julie M. Koeman,[6] Wubin Ding,[1] Manpreet Kalkat,[3] Nathan J. Spix,[3] Bryn Eagleson,[7] John Andrew Pospisilik,[3] Piroska E. Szabó,[3] Marisa S. Bartolomei,[8] Nicole A. Vander Schaaf,[3,10] Liang Kang,[3] Ashley K. Wiseman,[3] Peter A. Jones,[3] Connie M. Krawczyk,[5] Marie Adams,[6] Rishi Porecha,[4] Brian H. Chen,[9] Hui Shen,[3,*] and Peter W. Laird[3,11,*]

[1]Center for Computational and Genomic Medicine, Children's Hospital of Philadelphia, Philadelphia, PA 19104, USA
[2]Department of Pathology and Laboratory Medicine, University of Pennsylvania, Philadelphia, PA 19104, USA
[3]Department of Epigenetics, Van Andel Institute, Grand Rapids, MI 49503, USA
[4]Illumina, Inc., Bioinformatics and Instrument Software Department, San Diego, CA 92122, USA
[5]Department of Metabolism and Nutritional Programming, Van Andel Institute, Grand Rapids, MI 49503, USA
[6]Genomics Core, Van Andel Institute, Grand Rapids, MI 49503, USA
[7]Vivarium and Transgenics Core, Van Andel Institute, Grand Rapids, MI 49503, USA
[8]Department of Cell and Developmental Biology, Epigenetics Institute, University of Pennsylvania Perelman School of Medicine, Philadelphia, PA 19104, USA
[9]FOXO Technologies Inc., Minneapolis, MN 55402, USA
[10]Present address: Department of Biological Sciences, Olivet Nazarene University, Bourbonnais, IL 60914, USA
[11]Lead contact
*Correspondence: wanding.zhou@pennmedicine.upenn.edu (W.Z.), hui.shen@vai.org (H.S.), peter.laird@vai.org (P.W.L.)

## SUMMARY

We have developed a mouse DNA methylation array that contains 296,070 probes representing the diversity of mouse DNA methylation biology. We present a mouse methylation atlas as a rich reference resource of 1,239 DNA samples encompassing distinct tissues, strains, ages, sexes, and pathologies. We describe applications for comparative epigenomics, genomic imprinting, epigenetic inhibitors, patient-derived xenograft assessment, backcross tracing, and epigenetic clocks. We dissect DNA methylation processes associated with differentiation, aging, and tumorigenesis. Notably, we find that tissue-specific methylation signatures localize to binding sites for transcription factors controlling the corresponding tissue development. Age-associated hypermethylation is enriched at regions of Polycomb repression, while hypomethylation is enhanced at regions bound by cohesin complex members. *Apc^Min/+* polyp-associated hypermethylation affects enhancers regulating intestinal differentiation, while hypomethylation targets AP-1 binding sites. This Infinium Mouse Methylation BeadChip (version MM285) is widely accessible to the research community and will accelerate high-sample-throughput studies in this important model organism.

## INTRODUCTION

Cytosine-5 DNA methylation is the most commonly analyzed epigenetic mark in higher eukaryotes, occurring primarily in the sequence context 5′-CpG-3′ in vertebrates. In mammals, DNA methylation plays a role in consolidating epigenetic states, coordinating differentiation and development, and suppressing transcription of endogenous transposable elements, among others.[1–3] A wide range of technologies has been developed over the past few decades to interrogate DNA methylation states.[4–7] PCR-based strategies emphasize locus-specific detection sensitivity, while bisulfite sequencing-based approaches can provide base-pair resolution DNA methylation in-

formation and array-based methods excel in efficiently handling large numbers of samples.

The gold standard for deep genome-wide characterization of DNA methylation has been whole-genome bisulfite sequencing (WGBS).[8] However, this comprehensive approach is fairly cost and analysis intensive. Reduced representation bisulfite sequencing (RRBS)[9] has presented an excellent compromise between sample throughput and coverage, and has recently been extended to cover more of the genome.[10] Single-cell WGBS provides high-resolution DNA methylation profiling of heterogeneous samples but generally at lower genomic coverage than deep bulk WGBS.[11–13]

These sequencing-based approaches provide detailed profiles of the methylome, but sample throughput is constrained by cost and analytic complexity, despite efforts to address these limitations.[14] For large, population-based cancer genome and epigenome-wide association studies, Infinium BeadArrays[15] are the platform of choice for cost-effective, high-throughput DNA methylation characterization. This technology uses a set of predesigned probes to interrogate hundreds of thousands of CpG sites simultaneously, outputting a fractional methylation level as a β value for each CpG. Infinium DNA methylation arrays have unquestionably dominated DNA methylation profiling from the perspective of sample size, with Infinium methylation data for more than 160,000 human samples deposited in the Gene Expression Omnibus (GEO) (https://www.ncbi.nlm.nih.gov/geo/). Large-scale projects, including The Cancer Genome Atlas,[16] cover additional tens of thousands of samples, not deposited at GEO. These studies have collectively produced more than 1,200 publications in PubMed referencing Infinium DNA methylation arrays.

Infinium DNA methylation arrays provide several distinct advantages. First, automated sample processing allows for cost-effective analysis of hundreds of samples in parallel within days, yielding highly reproducible and robust results. Sequence-based protocols are less standardized and have a much higher experimental failure rate. Second, the CpG content on the arrays has been intentionally designed and validated to represent biologically relevant sites in the genome with reduced redundancy. Therefore, the informational value is high despite the relatively small fraction of genomic CpGs interrogated. Third, the β value output for each CpG dinucleotide represents a high-precision measurement of the fraction of methylated molecules for that site in the sample, something that is only achieved with very deep sequencing using WGBS.[15,17] Fourth, the bisulfite-specific hybridization accentuates the evaluation of fully converted molecules, whereas sequencing-based approaches require bioinformatic discrimination between biological CpH methylation and incomplete bisulfite conversion.[18] Fifth, the consistency of identical, well-annotated CpGs, analyzed across all samples, makes for a highly streamlined and efficient data analysis, which is an often-overlooked cost component. Sixth, the characterization of the same set of CpGs on all arrays greatly facilitates and improves cross-study comparison, validation, and extrapolation. Seventh, the analysis pipelines for array data are more mature and standardized than for WGBS.[19,20] Moreover, the hybridization fluorescence data can be used to extract other types of genomic information that can be obtained by WGBS, such as copy-number variations and genetic ancestry.[18,20–22]

Despite the extraordinary contribution of Infinium arrays to human DNA methylation studies, there have been no equivalent commercial arrays targeting any model organism developed to date. There have been sporadic attempts to apply the human Infinium arrays to primates[23–26] and mice.[27–29] However, only 1% of Infinium methylation EPIC probes are mappable to the mouse genome, which can lead to array scanning problems, in addition to the cost inefficiency and poor content utilization. A custom Infinium array has recently become available targeting 36,000 conserved CpGs in mammals.[30]

The demand for a DNA methylation microarray is particularly high for the mouse. Mice have long been used as an exceptional

organism to model human physiology and disease because of their ease of handling, rapid breeding, and diverse genetics, as well as the evolutionary conservation between primates and rodents.[31–34] In fact, mouse was among the vertebrate species for which cytosine-5 DNA methylation was first reported in 1962.[35] The first eukaryotic cytosine-5 DNA methyltransferase was cloned from the mouse.[36] Much of our understanding of the role of DNA methylation in mammals stems from mouse models.[37–42] Experimental mouse studies generate large numbers of tissue samples at lower cost, with greater ease and fewer restrictions than human studies, and they allow integration with functional genetic analyses.[43] Therefore, a cost-effective, high-sample-throughput DNA methylation array efficiently targeting the most biologically relevant sites in the mouse genome would provide a much-needed tool for the mouse model research community. Here, we present the rational systematic design and first implementation of a highly efficient, commercially available Infinium Mouse Methylation BeadChip (version MM285), targeting almost 300,000 CpGs in the mouse genome, and we demonstrate its high reliability, reproducibility, and utility. We profiled 1,239 DNA samples encompassing 26 tissue or cell types as a comprehensive resource for the research community. We also disentangle aging and tumor-associated changes using this powerful tool and demonstrate different types of differentiation block by DNA methylation associated with aging and tumorigenesis. Finally, we constructed and validated an epigenetic clock for the mouse, using the array content.

## RESULTS

### Array design targeting biologically relevant epigenetic features

We designed the Mouse Methylation BeadChip in three main steps (Figure 1A). In the first step, we considered all genomic CpGs in the mouse genome with all possible combinations of the probe DNA strand (Watson versus Crick) and post-bisulfite conversion strand (converted versus replicated daughter strand). We screened potential probe designs for probe sequence mapping, genetic polymorphism influence, probe hybridization, and extension efficiency (see STAR Methods). In the second step, we selected CpGs to target 13 genomic features and eight groups of genomic regions associated with known biology. To account for unknown biology, we included 28,011 random CpGs. Next, we chose the best probe design for each included CpG by balancing probe design strand, number of replicates, and Infinium probe chemistry. We also added several categories of non-CpG probes to culminate in a total of 296,070 probes on the Mouse Methylation BeadChip (Figure S1A).

Probe selection covers the vast majority of protein-coding and long non-coding RNA (lncRNA) genes (Figure 1B). We consider CpGs within 1,500 bp from either direction of the transcript's transcription start site (TSS) as associated with the promoter for that transcript. We were unable to design suitable probes for 1,064 protein-coding genes. These genes are over-represented by olfactory receptor, vomeronasal receptor, defensin, and Prame gene families which display a high degree of sequence polymorphism, constraining probe design. We included probes for pseudogene and microRNA (miRNA) TSSs,

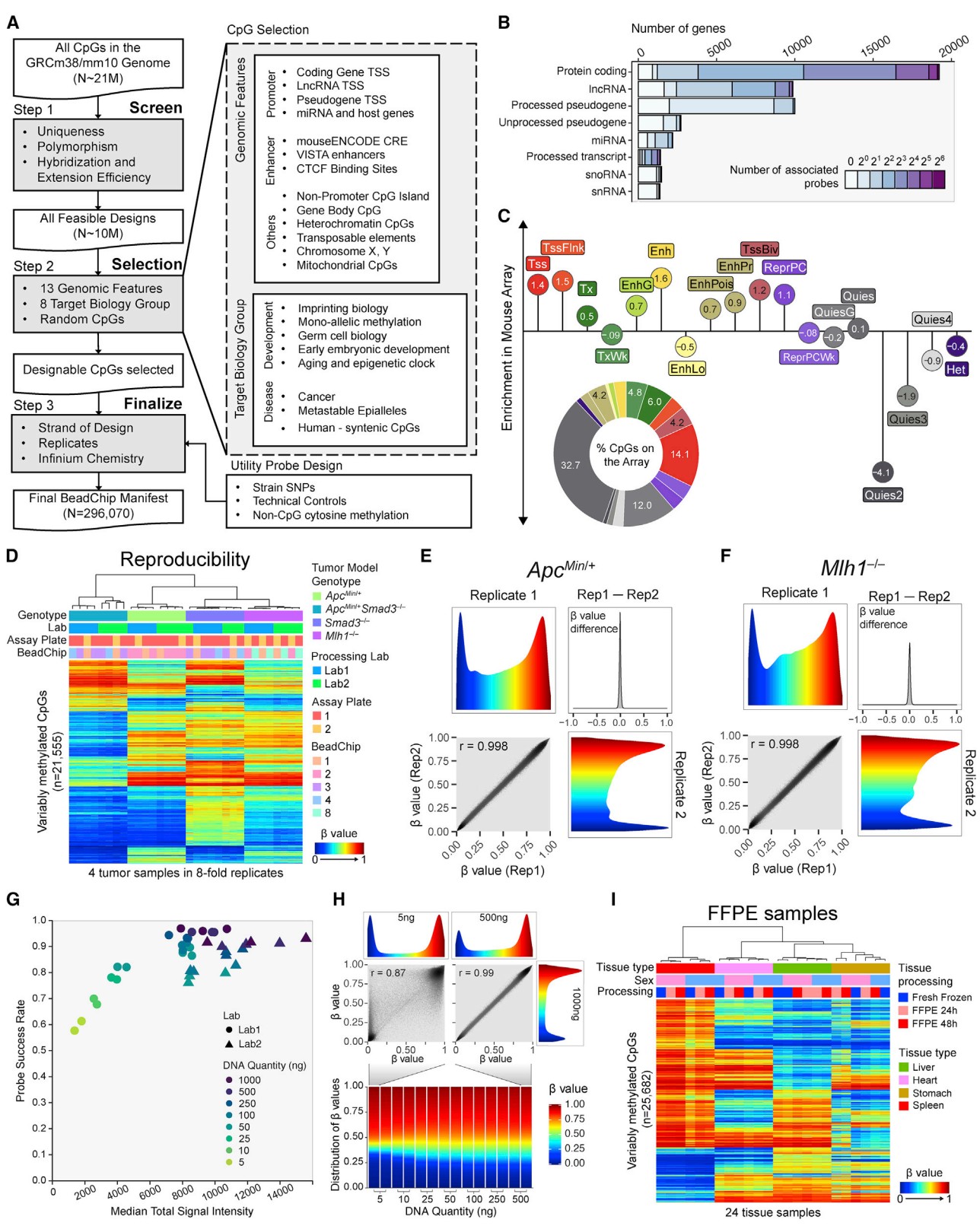

*(legend on next page)*

enhancers, and CTCF binding sites. We covered the majority of non-promoter CpG islands (CGIs), and included specific categories such as gene body CpGs, heterochromatic CpGs, transposable elements, sex chromosome-specific CpGs, imprinting control regions (ICRs), imprinting differentially methylated regions (DMRs), other mono-allelic methylation sites, CpGs relevant to germ-cell biology, early embryonic development, cancer, aging and epigenetic clocks, metastable epialleles, human syntenic regions, and mitochondrial CpGs, as well as non-CpG cytosine methylation probes (Figure S1A). We also added strain-specific SNP probes to facilitate strain confirmation and backcross tracing. Probe counts for the different design categories can be found in Figure S1A, with other basic parameters of the Mouse Methylation BeadChip, including probe mappability, bisulfite conversion strand designation, and probe redundancy (see STAR Methods), as well as comparisons with the human Infinium arrays in Figures S1B–S1I. Probes are indexed with an improved probe ID system to accommodate new features in probe design (see STAR Methods).

Because design-intent categories can yield probes associated with more than one such category, we also characterized the final array content by mapping each probe to one of several non-overlapping consensus chromatin states (Figure 1C). We derived consensus chromatin states from 66 ENCODE chromHMM calls[44] and plotted the abundance of probes on the array representing each chromatin category (Figure 1C, pie chart). Although quiescent chromatin probes represent a large fraction of the probes, normalization to the number of CpGs for each chromatin state in the entire genome reveals that enhancers and promoters are the most over-represented on the array compared with their relative abundance in the entire genome, underscoring the utility of the array in characterizing gene transcription control (Figures 1C [lollipop plot] and S2B). We observed a consistent representation of different chromatin states in the genome across different tissues (Figure S2C). The probes are distributed throughout the genome within both the euchromatic A and heterochromatic B compartments of the genome,[45] loosely corresponding to highly methylated domains and partially methylated domains (PMDs), respectively[8] (Figure S2D).

### High measurement reproducibility across experiments and laboratory settings
We profiled four colon tumor samples representing four different mouse tumor models with diverse DNA methylation profiles in 8-fold replicates to evaluate the technical reproducibility of the Infinium Mouse Methylation MM285 BeadChip. We employed two separate laboratories to evaluate the reproducibility of the complete experimental workflow, including bisulfite conversion, using identical 96-well plate layouts with replicates on different assay plates and BeadChip positions (Figure 1D). We have fully implemented data processing for the MM285 version of the mouse array in the SeSAMe (version 1.11+) Infinium DNA methylation data processing tool,[20] available on GitHub (https://github.com/zwdzwd/sesame) and Bioconductor (https://bioconductor.org/packages/release/bioc/html/sesame.html). All subsequent array analyses used SeSAMe data output. Unsupervised clustering of the 21,555 most variably methylated CpG probes revealed highly reproducible, tumor-specific DNA methylation profiles, with a negligible impact of laboratory facility or BeadChip position (Figure S2E). The between-lab Pearson correlation coefficient between technical replicates at the same plate and beadchip position was 0.9924 (n = 16 pairs) while the mean within-plate correlation coefficient between technical replicates at different beadchip positions was 0.9945 (n = 22 pairs), and the within-lab between-plate correlation at the same beadchip position was 0.9981 (n = 2 pairs). Representative replicate examples are shown in Figures 1E and 1F.

### Reproducible results with reduced input DNA quantities and FFPE samples
Although the recommended input DNA quantity for the Infinium Mouse Methylation BeadChip is 250–500 ng, we obtained excellent probe success rates (fraction of probes with the total fluorescent signal significantly exceeding background fluorescence) with input DNA quantities down to 100 ng (Figure 1G). We also obtained noisy, but usable, data with input DNA quantities as low as 5 ng (Figure 1H), consistent with other reports that Infinium BeadChip technology can be applied to as little as 10 ng of input DNA.[46] Low input DNA quantities resulted in a higher percentage of probe measurements masked by the pOOBAH approach for signal-to-background thresholding employed by the SeSAMe pipeline (Figure 1G).[20] As expected from the binary nature of DNA methylation, very low input DNA quantities resulted in a collapse of intermediate β values toward the extremes of the β distribution, with an accompanying reduction in correlation compared with high-quality data from larger DNA quantities (Pearson's r = 0.87) (Figure 1H).

---

**Figure 1. Mouse DNA methylation array design and technical validation**
(A) Workflow of designing the mouse Infinium methylation BeadChip.
(B) The number of probes associated with each gene model biotype.
(C) Enrichment of the probes in different chromatin states. The consensus chromatin states from 66 ENCODE ChromHMM files for twelve tissues were used. Distribution of interrogated CpGs in different chromatin states is shown at bottom left.
(D) Heatmap representing unsupervised clustering of four intestinal tumor samples, each analyzed in 8-fold replicate, including at two separate laboratory facilities. 21,643 most variable probes are shown.
(E and F) Smooth scatterplot of two replicates from *Apc*[Min] (E) or *Mlh1* (F) tumor samples. The colors represent DNA methylation β values, and the β value difference between the two replicates is shown on the top right quadrant.
(G) Probe success rate of samples of DNA input ranging from 5 to 1,000 ng. Data from two labs are represented by different shapes.
(H) Smooth scatterplots contrasting the comparison of a low-input sample (5 ng) and a high-input sample (500 ng), both with 1,000-ng-input sample. Samples are losing intermediate methylation reading as DNA input decreases.
(I) Reproducibility of DNA methylation measurement from four tissues comparing fresh frozen samples and samples fixed with formalin for 24 and 48 h.
See also Figures S1 and S2; Table S1.

We investigated the performance of the array on DNA samples extracted from formalin-fixed, paraffin-embedded (FFPE) tissues by allocating adjacent portions of mouse tissues to either fresh freezing, or 24- versus 48-h formalin fixation and paraffin embedding, prior to DNA extraction. We processed the DNA samples using the FFPE restoration kit (see STAR Methods). Unsupervised clustering with the most variably methylated probes revealed some evidence of minor sample processing effects on the data, but these were eclipsed by biological differences (Figures 1I and S2F). The mean correlation between matched FF and FFPE samples was 0.9875.

### Technical validation of the DNA methylation measurements using methylated DNA titrations and WGBS

To evaluate the accuracy of the DNA methylation measurements, we combined unmethylated DNA and M.SssI-treated fully methylated DNA (see STAR Methods), titrated to different mixing ratios from 0% to 100% methylated DNA (Figure 2A). The median β value of each sample corresponds well to the titrated percentage of methylated DNA (Figure 2B). To assess quantitative accuracy using an orthogonal method, we compared array results for the mouse B16 melanoma cell line DNA to WGBS of the same DNA sample. We observed a strong correlation between the two assay platforms for the vast majority of CpGs on the array (Pearson's r = 0.98) (Figure 2C). Taken together, these results confirm the ability of the mouse methylation array to measure the fraction of methylated molecules accurately and reproducibly for the vast majority of CpG sites interrogated.

### Biological validation of the DNA methylation measurements using *Dnmt1* knockouts and inhibitors

We assessed the impact of reduced levels of the main maintenance DNA methyltransferase, *Dnmt1*, on global DNA methylation distributions using gene-targeted embryonic stem cells (ESCs) (Figure 2D). We employed a series of hypomorphic alleles, $R$,[47] $P$, $H$,[48] and $N$,[38] as well as two null alleles, $S$[38] and $C$.[49] Heterozygous *Dnmt1* knockout ESCs did not show a clear reduction in DNA methylation levels, consistent with the presence of a remaining wild-type allele in the heterozygotes, but the median methylation of various homozygous and compound-heterozygous allelic combinations ranged from 20% to 55% of the average median of the wild-type lines (Figure 2D).

We have shown previously that combinations of *Dnmt1* hypomorphic alleles result in suppression of intestinal tumor formation in the $Apc^{Min/+}$ model of intestinal neoplasia commensurate with decreasing *Dnmt1* expression levels.[47] Here we show that global DNA methylation levels are proportionally reduced in colonic mucosa of mice with these allelic combinations (Figure 2E). The DNA methylation levels of diverse tissues in $Dnmt1^{N/R}$ mice also display proportionate decreases in DNA methylation compared with wild-type mice. The least methylated tissues in wild-type mice, such as muscle and stomach, are among the most hypomethylated tissues in $Dnmt1^{N/R}$ mice (Figure S3A). In contrast, testes are among the most highly methylated tissues in wild-type mice, and they also appear relatively resistant to hypomethylation except in the most severe hypomorphic $Dnmt1^{N/R}$ combination (Figures 2E and S3A). Studying

6,022 CpGs that are fully methylated in all tissues of the $Dnmt1^{+/+}$ mice, we observed that these regions were resistant to methylation loss in $Dnmt1^{N/R}$ mice across all tissues analyzed (Figure S3B). We found that these 6,022 CpGs were enriched 4.4-fold for transcribed regions compared with the overall mouse array content, suggesting that these may represent widely expressed genes that retain gene body methylation through Dnmt3b recruitment,[50,51] even in Dnmt1-deficient conditions. The extent of DNA hypomethylation at other regions in $Dnmt1^{N/R}$ mice varies by chromatin state (Figure S3C), suggesting a strong dependency of *Dnmt1* mutation-induced hypomethylation on genomic context. Our previous analyses had identified solo-WCGW CpGs at PMD regions as more susceptible to mitosis-associated methylation loss.[145] Consistent with these findings and Dnmt1's role as the primary maintenance methyltransferase, solo-WCGW CpGs showed the deepest methylation loss in $Dnmt1^{N/R}$ mice among all the chromatin states and CpG categories (Figures S3C and S3D).

The discovery 40 years ago of 5-aza-cytidine as an inhibitor of DNA methylation in mouse 10T1/2 fibroblasts helped to launch the modern era of DNA methylation research.[52] We repeated this experiment with 5-aza-2′-deoxycytidine (decitabine [DAC]) and observed substantial hypomethylation, with the median β value of DAC-treated cells reaching 81% of the median β value of mock-treated DNA at day 12, with some recovery by day 24 to 86% of mock-treated cells (Figure 2F), consistent with prior reports.[53,54]

### Methylation biology of genomic elements

We explored the genomic distribution of DNA methylation at various genomic elements in 1,119 DNA samples representing 26 different cell and tissue types. As expected, we observed low DNA methylation levels at promoter regions (Figure 3A). We did not identify such a dip at lncRNA promoters or miRNA promoters, consistent with a relatively repressed transcriptional state of most lncRNAs and miRNAs (Figure S4A). The integrated genomic position information across probes can be further leveraged to detect spatial DNA methylation patterns, such as the methylation states coupled to nucleosome phasing flanking CTCF binding sites (Figure 3A, leftmost panels).

Samples obtained from female mice displayed intermediate methylation levels at X-linked CGIs, consistent with X-inactivation in females but not in males (Figure 3B). We performed multivariate regression to identify sex-associated methylation differences and observed generally higher levels of methylation in females compared with males, attributable to the X-linked hypermethylation in females (Figure 3C). Notable regions of female-associated hypomethylation include X-linked non-coding RNAs, such as *Firre* and *Xist*, which are expressed on the inactive X chromosome and associated with the maintenance of its suppressive state (Figure 3C). A small number of genes on the X chromosome are known to escape X-inactivation.[55] We identified six X-linked genes predicted to escape X-inactivation with sufficient probe coverage on the array. We found that the methylation behavior of probes covering these genes in female colon samples was similar to that of male colon samples, consistent with escape from X-inactivation and distinct from that of most other X-linked genes (Figure S3E).

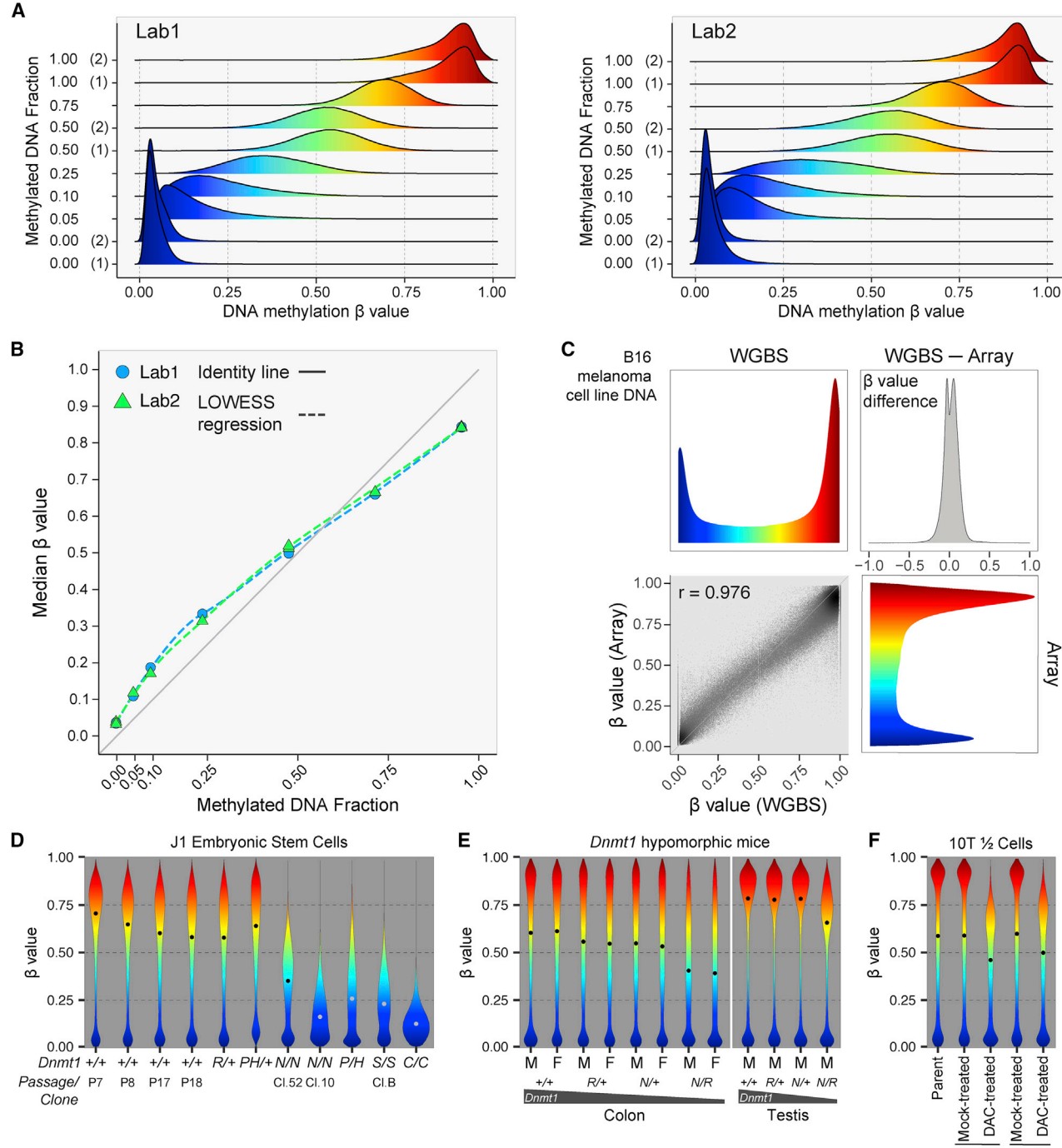

**Figure 2. Biological validation of the DNA methylation measurements**

(A) DNA methylation β value distribution of titration samples run from two different labs.

(B) Comparison of median DNA methylation β values measured from the Infinium Mouse Methylation MM285 BeadChip (y axis) with titration fraction of methylated DNA (x axis). Two different labs are shown in different colors.

(C) Comparison of DNA methylation β values measured from the MM285 array (y axis) with WGBS methylation level measurement on the mouse B16 melanoma cell line (45X mean CpG coverage) (x axis).

(D) Methylation level distribution of J1 embryonic stem cells with different *Dnmt1* genotypes.

(E) Methylation level distribution of colon and testis DNA from *Dnmt1* hypomorphic mice.

(F) Methylation level distribution of 10T1/2 cells treated with decitabine (DAC) at 200 nM, or mock-treated with PBS solvent for 24 h and then sampled 12 and 24 days after treatment.

See also Figure S3 and Table S2.

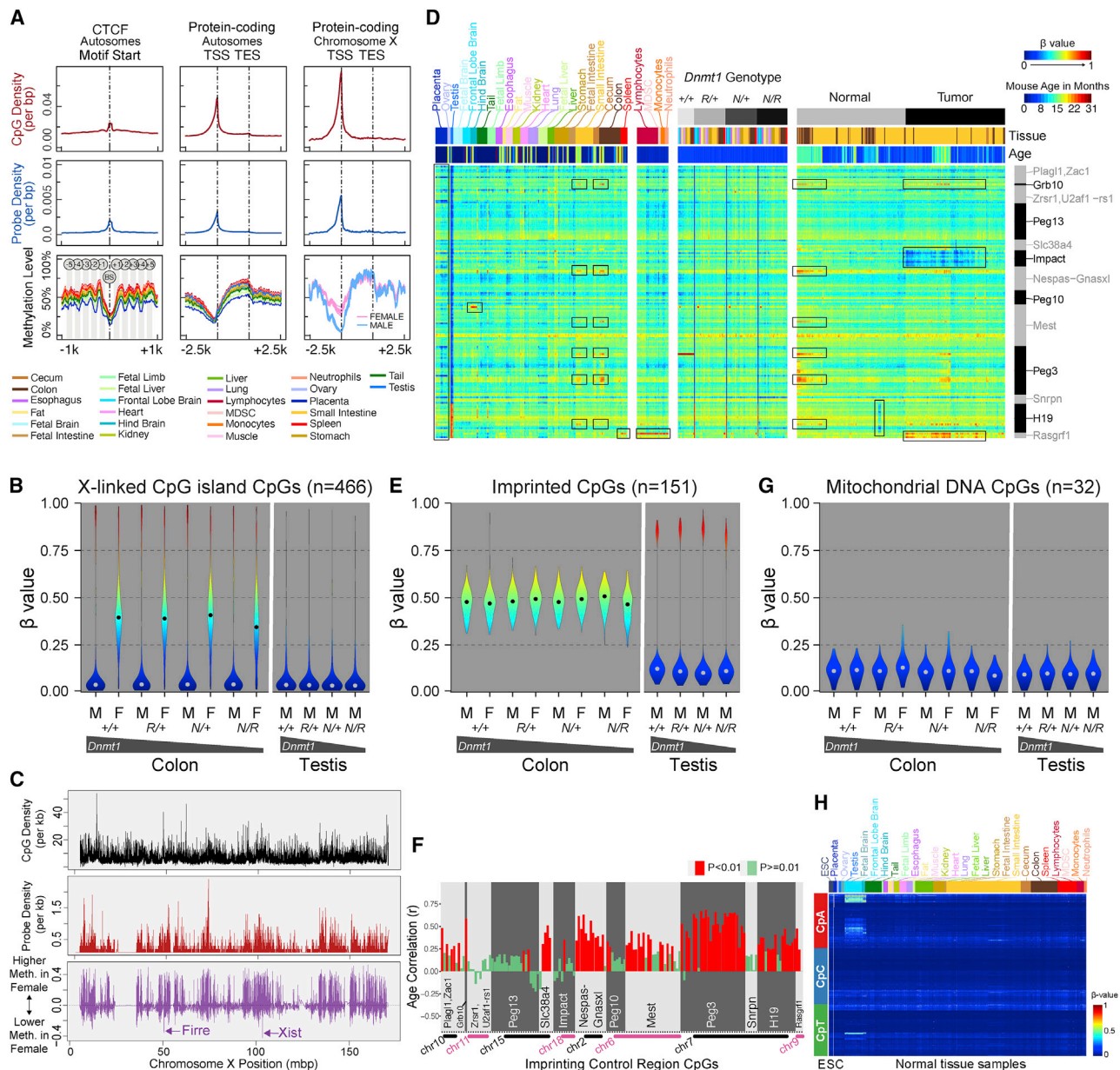

**Figure 3. Methylation biology of genomic elements**

(A) Genomic distribution of DNA methylation level centered on CTCF binding sites and protein-coding genes (autosomal and X-linked). The CpG density per bp (top row), the designed probe density per bp (middle row), and the average methylation level of samples stratified by tissue type (bottom row) are shown accordingly.

(B) Methylation level distribution of X-linked CpG-island CpGs in colon samples from male and female mice and testis samples from male mice.

(C) Genomic distribution of X-linked CpGs (top), mouse MM285 probe density (middle), and the sex-specific methylation difference, represented by the slope of the sex-specific effect in a multiple regression (bottom).

(D) Heatmap showing methylation level of CpGs (rows) associated with 13 curated imprinting control regions (ICRs) in non-malignant and intestinal tumor tissue samples (columns). CpGs are sorted by genomic coordinates. The associated ICR regions are labeled on the right. The grayscale horizontal bar on top represents Dnmt1 genotype and malignancy state. The second bar represents tissue type, using the color key as indicated above the left side of the heatmap. The third bar represents mouse age with the color key indicated on the top.

(E) DNA methylation level distribution of CpGs associated with ICRs or secondary differentially methylated regions. Colon (left) and testis (right) samples from mouse samples of two sexes and *Dnmt1* genotypes are shown.

(F) Correlation with age of CpG DNA methylation levels at 13 ICRs in 128 colon and small intestine tissue samples.

(G) Distribution of mitochondrial CpG methylation in colon and testis samples of different sex and *Dnmt1* genotype.

(H) Heatmap showing CpH methylation level. Rows correspond to cytosine loci in CpA, CpC, and CpT context. Columns correspond to ESCs and different tissue samples.

See also Figure S4; Tables S3 and S4.

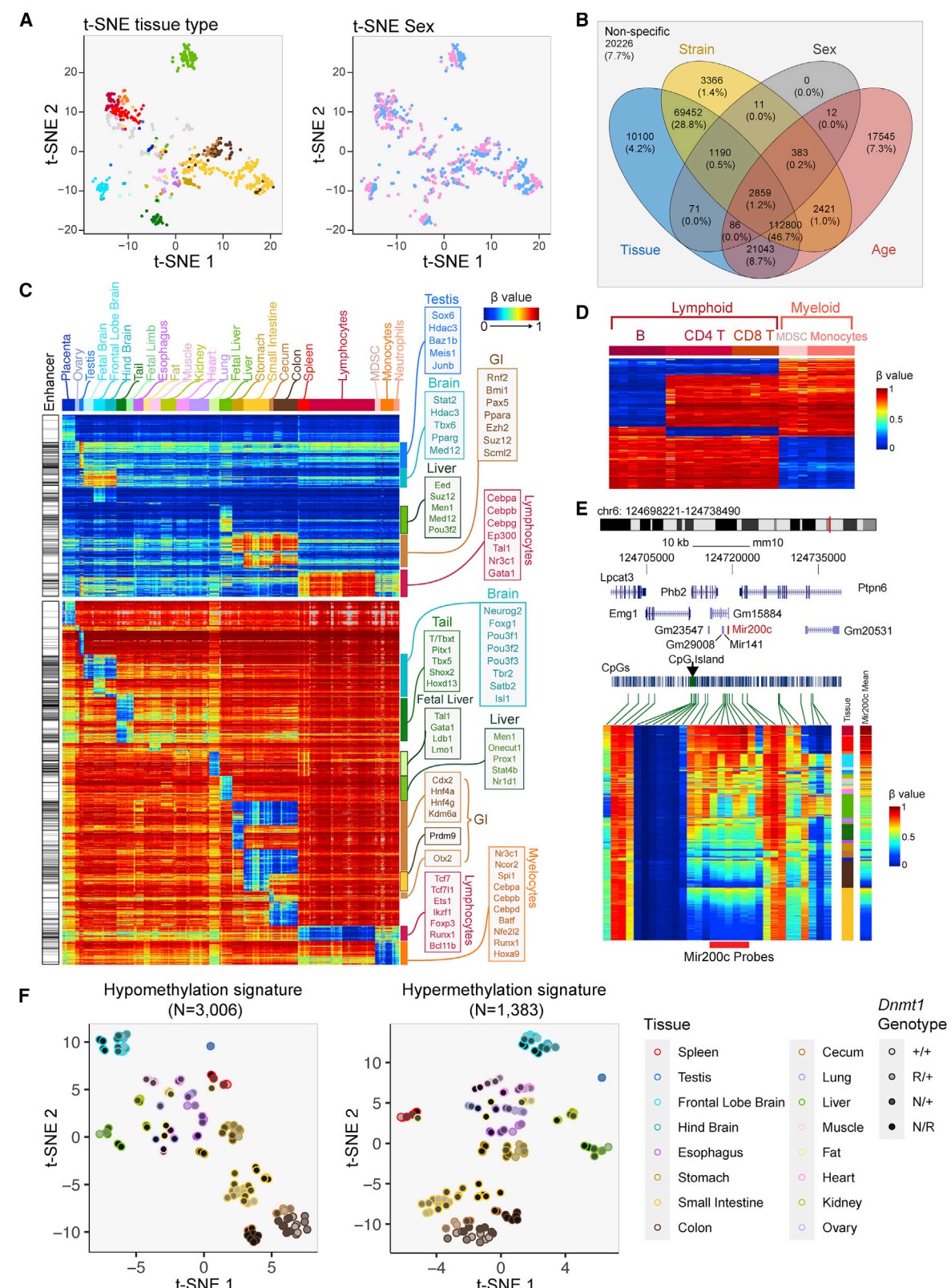

We confirmed that CpGs at ICRs and DMRs in imprinted regions display intermediate methylation levels, consistent with their mono-allelically methylated state (Figure 3D). In male germ cells, genomic imprints are erased and paternal imprinting patterns established, in some cases as a methylated site or more commonly as an unmethylated state at each individual locus, resulting in polarization of DNA methylation levels at most imprinted loci in the testis (Figures 3D and 3E, right). We analyzed the *H19-Igf2* ICRs and secondary DMRs in detail across mouse somatic tissue samples and verified intermediate DNA methylation for the corresponding CpGs (Figure S4B). We performed comprehensive annotation of probe association with previously reported mouse ICRs (see key resources table). We also performed *de novo* annotation of potential mono-allelically methylated and parent-of-origin-associated differential methylation based on DNA methylation levels in the soma and testes, using previously generated WGBS data on sperm and oocytes (see STAR Methods and Figure S4C). One hundred and fifty-one CpGs that are strictly intermediately methylated in somatic tissues fall in 13 previously reported ICRs (Figure S4C) (see STAR Methods for availability of these probe annotations).

Most CpGs associated with the 13 curated ICRs displayed consistent intermediate methylation levels (Figure 3D). However, a subset of these CpGs revealed tissue-specific or tumor-associated loss of the intermediate methylation (LOIM) patterns (framed by boxes). Notably, placental tissues display LOIM in a subset of CpGs across multiple ICRs. One *Rasgrf1*-associated CpG (cg47072751_BC21) displayed a gain of methylation only in the hematopoietic cells, as seen in sorted blood cells and spleen tissue, largely composed of blood cells. One CpG in the *Mest* ICR revealed gain of methylation in a hindbrain-specific manner. Intriguingly, we identified a coordinated LOIM at CpGs across multiple ICRs in tissues of the gastrointestinal tract, representing primarily a gain of methylation. We found a positive association of this methylation gain with age (Figures 3F and S4D). In intestinal tumors, we observed regional hypomethylation of the *Impact* ICRs, whereas *Rasgrf1* displayed regional hypermethylation (Figure 3D, right).

In colon tissues with hypomorphic Dnmt1 alleles, imprinted regions (Figure 3E, left) appeared to resist loss of methylation compared with other genomic regions (Figure 2E, left). We observed varying susceptibility among imprinted regions to hypomethylation in *Dnmt1* gene-targeted ESCs with more severe global hypomethylation (Figure S4E). Among imprinted regions, *Rasgrf1* was the most resistant to hypomethylation, followed by *Peg13* and *H19*. Other imprinted loci appeared more susceptible to loss of methylation in *Dnmt1* hypomorphic ESCs.

We incorporated probes in the mouse array design for metastable alleles and variably methylated regions (VMRs) that had been described in the literature.[56–59] These regions were described as varying in methylation levels across individuals of the same age, strain, and sex, within a defined cell type or tissue context. However, we observed that the VMR probes were enriched for CpGs displaying tissue-, strain-, and sex-associated differential methylation (Figure S4F). Therefore, it was important to investigate whether these CpGs displayed more variable methylation if we constrained our analysis within a fixed cell type, strain, and age. We analyzed the methylation variability at VMR probes in flow-sorted B cell samples of the same age and strain, and did indeed observe significantly higher variance of methylation levels across individual mice (Figure S4G, left panel). However, we also noted that VMR probes have more intermediate mean β values (Figure S4G, right panel), allowing for a higher variance on the β distribution scale.[60] We did not observe a significant difference in variance between VMR and non-VMR probes with a mean β value constrained between 0.5 and 0.7, suggesting that intermediate methylation values are a major determinant of variable methylation at VMRs.

Mitochondrial CpGs appeared mostly unmethylated, as expected, owing to a lack of access to nuclear DNA methyltransferases (Figure 3G). We assessed non-CpG (CpH) methylation in 798 samples from diverse tissues, using 2,310 CpH probes included in the array design. Most tissues revealed very low levels of CpH methylation (Figure 3H), consistent with the literature. We identified moderate levels of CpH methylation, particularly CpA methylation, in hindbrain and frontal lobe brain samples in adult mice (Figure 3H). This CpH methylation is absent from fetal brain samples, consistent with the establishment of CpH methylation during postnatal development, whereas CpG methylation is established during early development.[61]

## Identification of tissue-specific DNA methylation patterns and signatures

We explored the dominant factors influencing variations in the mouse methylome by projecting each of the 1,076 samples to two-dimensional space, using t-distributed stochastic neighbor embedding (t-SNE) applied to all methylation probes on the array (Figure 4A). Coloring the samples by different meta-labels reveals the extent to which the DNA methylome is driven by different factors. Sample clustering was dominated by tissue type (Figure 4A, left). We further clustered these DNA methylomes using DBSCAN and calculated the uncertainty coefficient using cluster membership and sample meta-information. Consistent with the t-SNE analysis, tissue type is the dominating

---

**Figure 4. Identification of tissue-specific DNA methylation patterns and signatures**
(A) t-SNE cluster map showing samples clustered by tissue type (left) and sex (right). The color legend is the same as in Figure 3A.
(B) Four-way Venn diagram showing CpGs differentially methylated with respect to strain, tissue, sex, and age in 467 mouse samples.
(C) CpGs (rows) specifically methylated (top) and unmethylated (bottom) in each tissue (columns showing samples organized by tissue type).
(D) CpGs (rows) specifically methylated (top) and unmethylated (bottom) in each sorted leukocyte cell type (columns showing samples organized by cell type).
(E) Track view of the DNA methylation profile of the Mir200c locus as a marker for epithelial versus mesenchymal cells. Tissue type and mean methylation level of five CpGs associated with the Mir200c locus (shown by the red bar at the bottom) are shown on the right-hand side of the heatmap.
(F) t-SNE cluster map of primary tissue samples from wild-type and *Dnmt1*[NR] mice using tissue-specific hypomethylation (left) and hypermethylation signature (right). The color legend is the same as in Figure 3A.
See also Figure S5.

factor of DNA methylome determination, followed by age, strain, and sex (Figures 4A [right], S5A, and S5B). Hierarchical clustering of 246 samples representing 22 primary tissue types revealed grouping primarily by tissue type (Figure S5C), verifying the importance of tissue type in methylome determination. We then performed whole-array multivariate regression analysis of CpG methylation on tissue, strain, sex, and age on 467 non-tumor samples (Figure 4B). The number of CpGs showing significant DNA methylation differences associated with each variable and their interactions are shown in a Venn diagram. Consistent with our unsupervised analysis, tissue type and age were the strongest individual drivers of specific methylation behavior. Many CpGs displayed joint influences from multiple covariates on methylation levels. For example, methylation at 95% of the tissue-specific CpGs is under joint influence by other factors, including strain (Figure 4B).

In light of the tissue specificity of methylation levels at many CpGs on the array and the value of methylation-based tissue deconvolution in various experimental settings, we sought to identify a panel of probes that were differentially methylated in specific tissues (Figure 4C). Grouping samples by tissue types, Figure 4C represents probes (rows) that are specifically hyper- or hypomethylated in each tissue (columns) compared with the rest of the tissues. Some methylation signatures that are shared across related tissue types (e.g., tissues from the gastrointestinal tract) are also indicated. Certain tissue types are associated with more abundant tissue-specific methylation than others (e.g., fat, kidney, lung). These tissue-specific methylation signatures are associated with binding of transcription factors controlling the development of the corresponding tissue (Figures 4C, S5D, and S5E).

The observation of more CpGs associated with tissue-specific hypomethylation compared with hypermethylation reflects the preference for transcription factors to bind unmethylated DNA.[62] This preference for methylated or unmethylated DNA leads to tissue-specific transcription factors being divided into two groups. For example, lymphocyte-specific hypomethylation is associated with *Tcf7*, *Ets1*, *Ikzf1*, and *Foxp3* while the lymphocyte-specific hypermethylation is associated with *Cebpa/b/g*, *Tal1*, and *Gata1*. Interestingly, *Cebpa/b/g* is associated with lymphocyte-specific hypermethylation but, on the other hand, myelocyte-specific hypomethylation, revealing a complex mode of transcription factor-DNA interaction.[63] We used fluorescence-activated cell sorted leukocytes to further distinguish B cells, CD4 T cells, CD8 T cells, and monocytes (Figure 4D). These methylation signatures can be used to investigate the presence or absence of cell types in a mixed tissue/cell type scenario.

Methylation of the Mir200c locus is a known signature of human mesenchymal cells.[64,65] We evaluated CpGs located in the genomic neighborhood of the Mir200c gene and identified five CpGs whose methylation can reliably discriminate mesenchymal cells from epithelial cells. Tissues composed of largely epithelial cells such as colon and small intestine are largely unmethylated, while tissues that consist primarily of mesenchymal cells such as leukocytes are mostly methylated at this locus (Figure 4E). Using these probes, we can derive a DNA methylation-based estimator of tissue purity for epithelial tumors, which we utilized in tumor studies shown in Figure 7.

As these tissue-specific methylation differences can be used to mark tissue/cell identity, we next investigated to what extent altering the methylome by *Dnmt1* hypomorphic alleles would impact the methylation of these sites and, potentially, cell identity. Although *Dnmt1* deficiency does cause some perturbation in these tissue-associated hypo- and hypermethylation signatures, the samples remain largely grouped by tissue type in t-SNE (Figure 4F). Visual inspection in an ordered heatmap reveals that the tissue specificity of these signatures is retained well (Figure S5F), consistent with the preservation of observed tissue anatomy and histology.

### Inter-species comparisons and applications to patient-derived xenografts

We explored the utility of this array for other mammalian species. We mapped the array probes to 310 genome assemblies, as collected in Ensembl (version 101). Figure 5A depicts the number of mappable probes from 56 species for our array and in comparison with the human Infinium DNA methylation EPIC array. As expected, the number of functional probes decreases as evolutionary distance increases. The Infinium Mouse Methylation BeadChip is predicted to function optimally in *Mus musculus* but should also be applicable to related species such as Algerian mice, Ryukyu mice, and Shrew mice. We experimentally investigated the performance of the mouse array in human samples as well as in other rodents. We assessed performance on human, rat, mouse, and hamster (CHO cell line) DNA. The observed signal intensities in each probe category are consistent with the predicted probe annotations (Figure S6A). Using 13,962 probes with high mapping quality to the rat genome, we experimentally validated that these probes faithfully represent varying methylation levels in rat DNA, using mixtures of fully methylated and unmethylated rat DNA titrated at varying ratios (Figure 5B).

To investigate the cross-application of the human and mouse arrays, we identified probe sets predicted to work for both species on both human (EPIC) and mouse (MM285) arrays, as well as probes that are predicted to only work for the designed species. We experimentally tested these using mixtures of human and mouse DNAs at varying ratios analyzed on both the human (EPIC) and mouse methylation arrays (Figure 5C). As expected, the probe sets that are predicted to work for both species maintain a high probe success rate as the ratios of mouse versus human DNA varies. In contrast, probes that were designed for one species show a reduction in probe success rate as the fraction of other species' DNA increases (Figure 5C). Since the Infinium platform can accommodate low input DNA quantities (Figure 1H), the presence of non-hybridizing DNA from another species does not become problematic until an overwhelming excess of the other species is reached.

The study of evolutionary conservation of DNA methylation requires proper control of other biological covariates such as tissue. To study the conservation of methylation patterns and investigate how tissue type may impact this conservation, we first performed principal component analysis using probes targeting CpGs in synteny with EPIC CpGs (Figure 5D). As expected, CpGs designed to map to EPIC CpGs are more evolutionarily conserved than the other nuclear DNA CpG categories (Figure S6B). The first principal component (PC1) is strongly

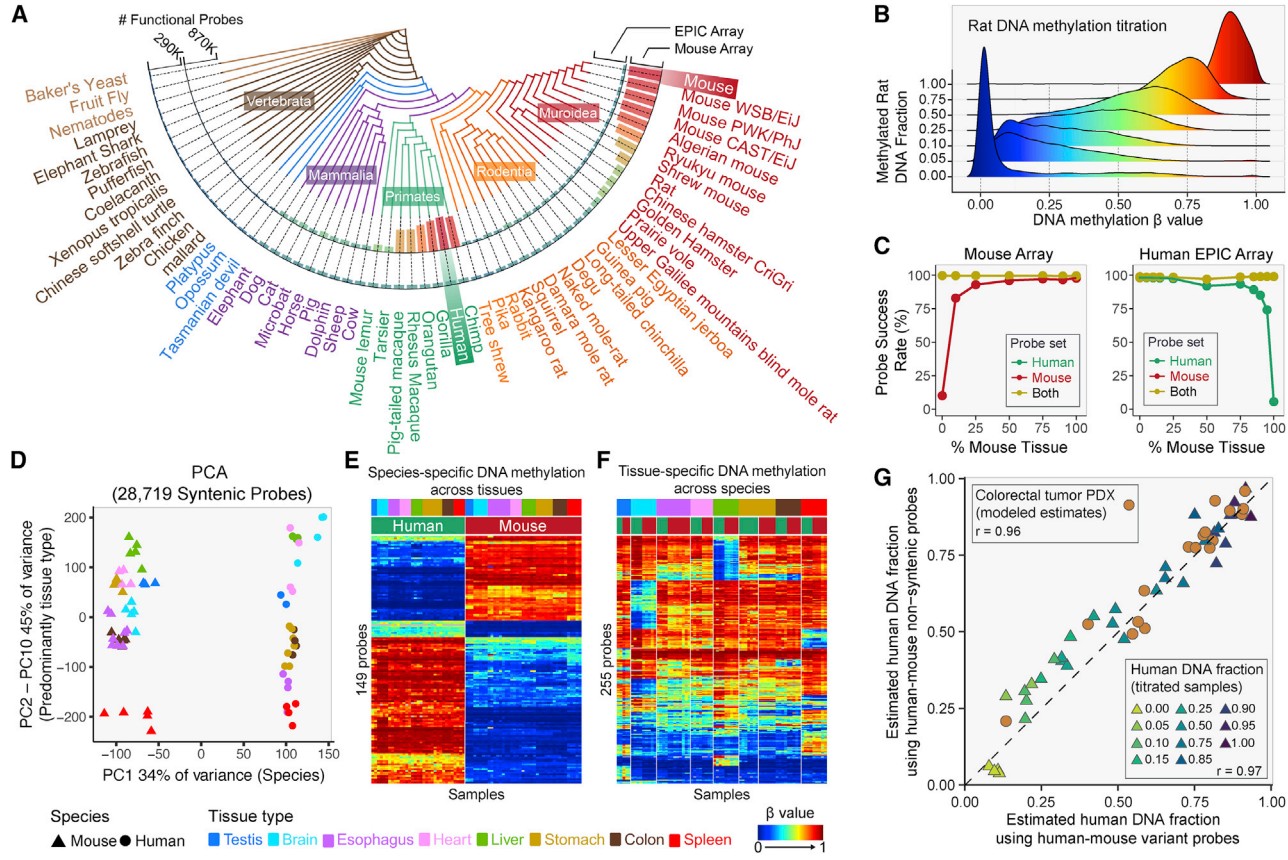

**Figure 5. Inter-species comparisons and applications to patient-derived xenografts**

(A) The numbers of functional EPIC and MM285 array probes on different vertebrate and invertebrate species.

(B) Distribution of the DNA methylation reading of functional probes in the rat genome. Rat DNA samples were derived from mixing fully methylated and fully unmethylated rat DNA using titration ratios from 0% to 100%.

(C) Probe success rates of the mouse (left) and EPIC arrays (right) on human-mouse mixture titrated at a range of percentages of the mouse DNA (x axis). Curves of different color represent probe sets that work on either both species (yellow) or only the designed species.

(D) Principal component analysis showing the leading PC (PC1) on the x axis and combined PC2-10 on the y axis. Samples are colored by tissue with symbol shape representing species.

(E and F) Heatmap display of the DNA methylation level of species-specific probes (E) and tissue-specific probes (F).

(G) Cross-validation of estimating human DNA fraction using human-mouse variant probes (x axis) and using human-mouse non-syntenic probes from both mouse and EPIC arrays. Each triangle represents a human-mouse titration sample. Each circle represents a PDX sample. Color of the triangle corresponds to the known titration fraction of human DNA.

See also Figure S6; Tables S5 and S6.

associated with a species-specific effect (Figure S6C), accounting for 34% of the methylation variation at syntenic CpGs. Pairwise correlation between human and mouse tissues in the syntenic CpG methylomes revealed a varying degree of tissue correspondence: the mouse esophagus, stomach, heart, spleen, and testis are best correlated with the human corresponding tissue (Figure S6D). In contrast, mouse brain, liver, and colon are not well correlated with their human counterparts. We next regressed the methylation level of human-mouse syntenic CpGs on both tissue and species. We identified extensive interaction of tissue and species effect in the regression analysis, leaving only 149 syntenic CpGs showing species-specific methylation but no tissue-specific methylation (Figures 5E and S6E), and only 255 CpGs displaying only tissue-specific methylation (Figures 5F and S6E).

The color channels for Infinium-I probes are determined by the identities of the probes' extension bases.[18] A subset of the probes that map perfectly to the human genome can be associated with switched color channel due to single-nucleotide variations (SNVs) between human and mouse at the probe extension base. These probes emit signal in one color channel when hybridized to human DNA and the other when hybridized to mouse DNA. We identified 19 such human-mouse variant probes on the mouse MM285 array. Using these probes, we can measure the fraction of human DNA in human-mouse mixture samples such as patient-derived xenografts (PDXs). We can compare this method with the total fluorescent signal intensities for species-specific non-syntenic probes. A comparison between these two approaches is shown in Figure 5G for mixed DNA samples of human and mouse DNA titrated at known varying ratios and

analyzed on both the mouse and human (EPIC) DNA methylation arrays (Figures 5G and S6F–S6I). The two methods for estimating the fraction of human DNA correspond well with each other (r = 0.97). Given the success of this analysis, we then went on to estimate the fraction of human DNA in 17 colorectal cancer PDX samples, using both methods to model estimates of human DNA fraction in these xenografts (see STAR Methods). The two methods provided highly correlated estimates in PDX samples (r = 0.96, p < 0.05) (Figure 5G).

### Mouse-strain-specific SNP probes facilitate tracing of backcross generations

To facilitate epigenetic studies using inbred mice, we included 1,485 SNP probes for strain genotyping in the mouse MM285 array design. These probes were selected from a curated phylogeny of common inbred mouse strains, including C57-related mice and Castle mice as well as some common wild mice (Figure 6A). We retrieved the whole-genome sequence data from the Mouse Genome Project for each of the included strains and identified SNVs that represent the segregation of each clade on the phylogeny (see STAR Methods). We extracted DNA from various tissue types from 25 different mouse strains and found that these probes can be used to uniquely identify strains in a cohort of 74 mouse DNA samples from 25 strains (Figure 6B). Note that some SNPs are shared across multiple mouse strains and represent clade-specific SNPs that can be used to study the genotype of mouse strains not included in the original collection of mouse strains. For example, we included SNPs that are common to all 129 strains in our collection, and these SNPs are thought to define the entire 129 clade and may be used to identify other 129 strains (Figure 6B). To illustrate the use of these SNP probes, we backcrossed 129S4/SvJaeJ mice to C57BL/6 mice for ten generations and extracted DNA at each backcross step. Using 28 SNPs that segregate the two strains, starting with pure inbred 129S4/SvJaeJ allelic homozygosity at these 28 SNPs, we then observe a shift at all loci to the heterozygous state, as expected for the F1 generation, and subsequently a progressive transition to C57BL/6 allelic homozygosity at all the SNPs tracked (Figure 6C). This result illustrates the utility of the Infinium Mouse Methylation BeadChip to identify mice with a mixed genetic background and to trace backcrossing progress. We developed a maximum-likelihood-based classifier that infers the closest strain information based on these SNPs. The prediction based on this classifier is highly consistent with the reported strain and can detect mice with a mixed genetic background (Figure S7A).

### Hypermethylation of Polycomb targets during aging

A major strength of the Infinium DNA methylation array platform is that it can be scaled to a very large number of diverse samples and thus enable us to distinguish epigenetic states specifically associated with potentially confounded traits, such as age and disease, as well as their interactions. The strongest predictor of cancer risk is age, so it can be challenging to distinguish between epigenetic alterations associated with tumorigenesis versus age or cellular proliferation. Inbred mouse models provide an opportunity to investigate these types of interactions in controlled experimental conditions with homogeneous genetic

backgrounds. We profiled the methylomes of 56 independent small intestinal tumor samples from $Apc^{Min/+}$ mice and performed multivariate regression of DNA methylation β values using age and tumor state as predictors to disentangle the relative contributions of tumor-associated versus age-associated DNA hypermethylation. We then conducted a comprehensive survey of the overlap between DNA protein binding elements[66] and either the tumor- or age-associated hypermethylation sites.

We found that age-associated hypermethylation occurs primarily at sites occupied by Polycomb Group (PcG) repressor complexes in intestinal cells[67] (Figure 7A, top), with a strong overlap with binding sites for both PRC1, such as *Cbx7* and *Rnf2*, and for PRC2, including *Ezh2* and *Suz12* (Figure 7B). The overlap of age-associated loss of methylation revealed an enrichment of binding sites for proteins that regulate chromatin loop stability, sister chromatid cohesion, and double-strand break repair,[68,69] including *Ctcf* and cohesin complex members *Smc1a*, *Stag1*, *Satb1*, and *Rad21* (*Scc1*) (Figure 7C), although the odds ratios were considerably lower than for the hypermethylation overlap.

### Hypermethylation of differentiation regulatory elements in intestinal tumorigenesis

We found that CpGs displaying tumor-associated hypermethylation are enriched for regions containing enhancer elements (Figure 7A, bottom). Tumor-associated hypermethylation was associated with transcription factors implicated in intestinal tissue development, best exemplified by homeodomain transcription factors *Cdx2* and *Prox1*, and nuclear factors such as *Hnf4a*, *Hnf4g*, *Vdr*, and *Nr2d2* (Figure 7D).[70] An analysis of the associated target genes of these transcription factors indicates an enrichment of gene ontology terms involved in intestinal cell terminal differentiation and function (Figure 7E). We hypothesize that hypermethylation of transcription factor binding sites controlling enterocyte function results in a differentiation block, which could promote tumorigenesis. Tumor-associated hypomethylation was associated with binding sites for the AP-1 transcription factor (Figure 7F), consistent with our previous report on human colorectal cancer.[71]

### Construction of a mouse epigenetic clock

There has been broad interest in the application of DNA methylation profiles as a predictor of chronological age.[72–74] Several epigenetic clocks have been established for the mouse, largely using RRBS.[75–79] We collected 706 mouse tissues and constructed an epigenetic clock for the MM285 array using an elastic net framework, selecting for 347 CpGs that predict the chronological age of the mouse with a mean absolute error of 1.2 months (see STAR Methods) (Figure 7G). Although the cellular expansion associated with tumorigenesis might be expected to distort the biological age of a tissue, we found in an analysis of 156 intestinal tumor samples that the clock prediction from our normal tissue model agreed with the reported chronological age of mice bearing these tumors, but with a greater absolute mean deviation (Figure 7H). The clock features selected in the model are composed of CpGs that both gain and lose methylation with age (Figure S7B) and CpGs associated with Polycomb-targeted chromatin (Figure S7C). Among tissue

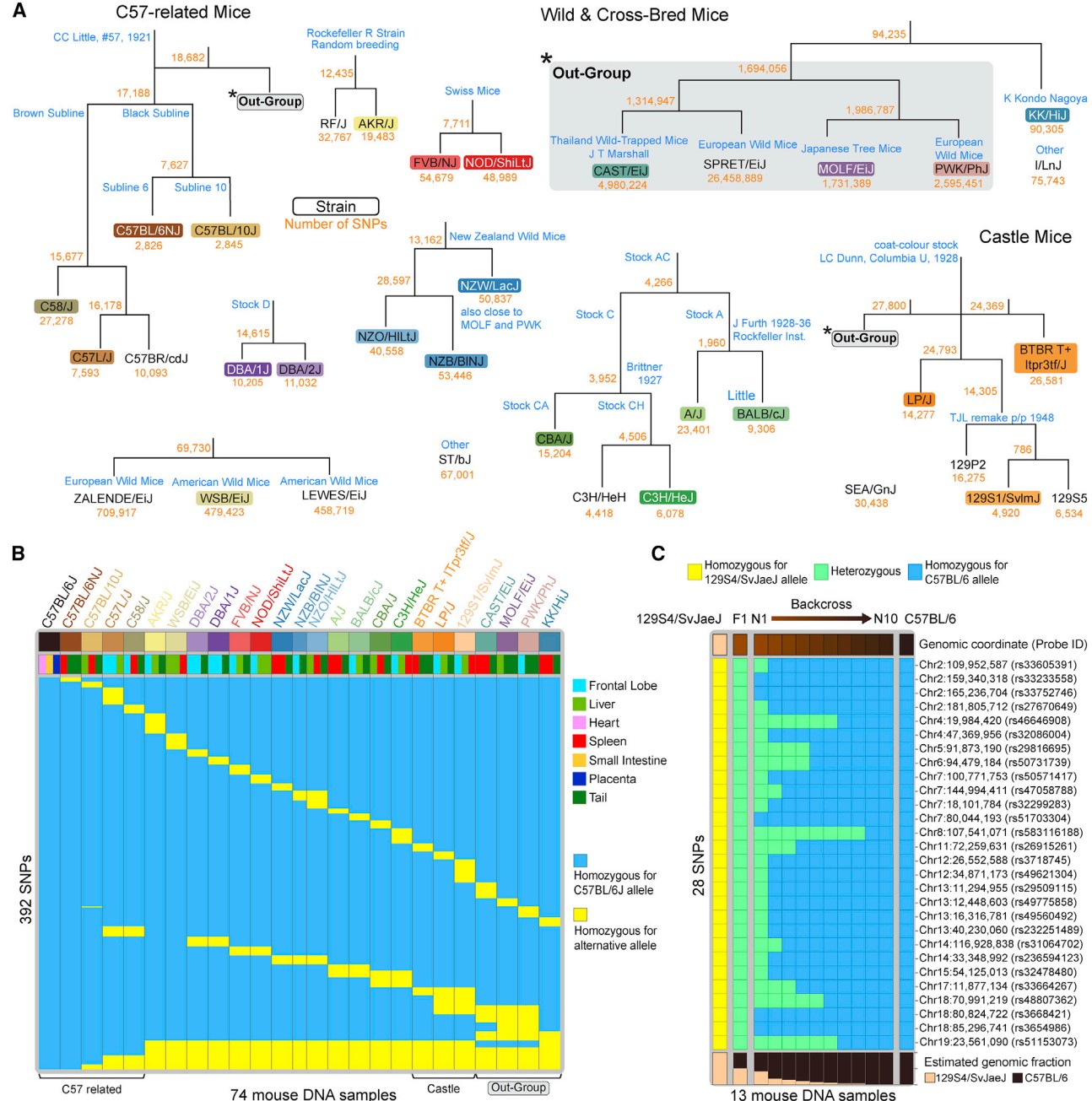

**Figure 6. Mouse strain-specific SNP probes facilitate tracing of backcross generations**

(A) Inbred mouse strain pedigree and SNP probe design. Numbers in yellow indicate the number of segregating SNPs in the whole genome, from which SNP probes were selected.

(B) Genotyping result of 27 mouse strains. Rows correspond to designed SNP probes. Columns correspond to samples. Rows are ordered such that probes that uniquely isolate a mouse strain are listed on the top, followed by probes that identify multiple strains.

(C) Strain genotyping on multiple generations of the 129Sv mice backcrossed to C57BL/6 mice. Each row corresponds to a 129/Sv-C57BL/6 segregating SNP. The fraction of 129/Sv allele is shown at the bottom.

See also Figure S7.

types, testis displays the highest age overestimate, although this is based on just three samples (Figure S7D). Its epigenome is associated with the unique developmental trajectory of male germ cells.

**DISCUSSION**

Mouse models have profoundly impacted basic and translational biological research, contributing to many important discoveries

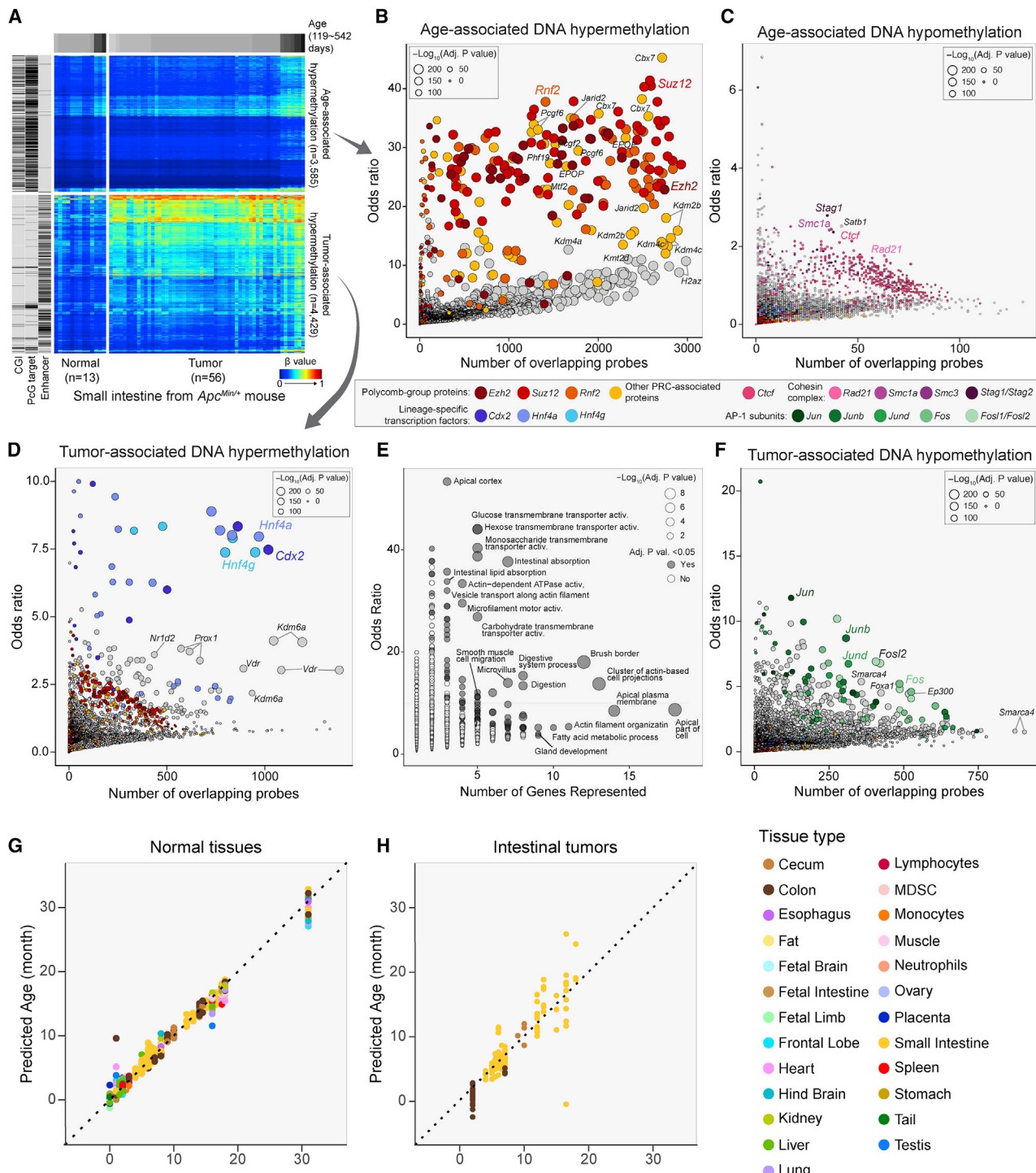

**Figure 7. Age- and tumor-associated DNA methylation and the development of an epigenetic clock**

(A) Heatmap displaying DNA methylation gain associated with aging (top) and with tumorigenesis (bottom). Rows correspond to CpGs and columns correspond to samples. Left-hand side bar represents whether probes (rows) fall to CpG islands (CGI), Polycomb repressive complex group (PcG) target, and enhancer elements in small intestine samples.

(B) Transcription factors whose binding sites are enriched in CpGs that gain methylation with age. y axis represents odds ratio of enrichment, and x axis represents number of significant probes overlapping binding sites. Size of the dots denotes statistical significance of the enrichment (Fisher's exact test).

(C) Transcription factors whose binding sites are enriched in CpGs that lose methylation with age.

*(legend continued on next page)*

in the past century, including the elucidation of basic principles of epigenetics.[80–88] While our understanding of DNA methylation dynamics in genomic and chromatin contexts has benefited greatly from bisulfite sequencing studies of limited sample size in both mice and humans, large population-based DNA methylation studies and cancer genome projects continue to rely on cost-effective Infinium arrays, which excel at sample throughput, reliability and reproducibility, consistent feature sets, and analytic convenience. These features of Infinium arrays constitute a highly efficient and effective DNA methylation profiling tool for the large sample sizes dictated by the genetic heterogeneity of human populations and the diversity of human pathologies. With profiles of more than 160,000 healthy and diseased tissue specimens publicly available, this platform has clearly dominated large-scale DNA methylation studies in humans. The widespread use of mouse models in mechanistic and preclinical studies of human disease, drug development, embryology, immunology, nutrition, toxicology, metabolism, and numerous others,[89–93] together with the tremendous power of classical and molecular mouse genetics,[31,32] results in tens of millions of mice used in research annually.[94,95] Our understanding of the role of epigenetics in many of these systems has been stymied by the lack of a widely accessible, high-sample-throughput DNA methylation profiling method for mice. Here we describe the design, validation, and application of such a tool, and present a dataset of 1,239 DNA samples representing a comprehensive DNA methylation atlas for the mouse.

We undertook the design of this mouse MM285 array with the goal of representing the known diversity of mouse methylation biology as efficiently as possible, while adding other genomic elements of interest with the potential for methylation variation and allowing for the discovery of unknown biology through the inclusion of randomly selected CpGs. The high degree of local methylation correlation among CpGs[96] provided us with some flexibility to select probes with high technical design scores for representative regions. We covered all designable known promoters for coding and non-coding transcripts, enhancers, CGIs, CTCF binding sites, imprinted loci, epigenetic clock loci, and CpH methylation. We included all CpGs syntenic with the human Infinium EPIC array to maximize translation to human biology. We took advantage of publicly available WGBS datasets to prioritize probes with known biology, as well as the opportunity to include other types of useful probes such as strain-specific SNPs. We also randomly selected CpGs to enable discovery of novel biology in the event of limited mouse WGBS data, e.g., colon cancer-associated datasets.

The human Infinium DNA methylation array platform has benefited from more than a decade of technology optimization.[15] We incorporated this design knowledge in selecting optimal probes for the Infinium Mouse Methylation MM285 BeadChip.

As a consequence, the array performed extremely well in terms of reproducibility of methylation measurements, even between those obtained in different laboratory settings. We achieved excellent results with DNA extracted from FFPE tissues and with input DNA quantities as low as 5 ng, corresponding to less than 1,000 cells. Most commonly used WGBS methods require at least 100 ng of DNA input, although some can be pushed down to 10 ng.[97] Standard RRBS requires a comparable 10–200 ng DNA input.[98] Nevertheless, as template molecule quantities decrease, allelic dropout increases during whole-genome amplification, resulting in a loss of intermediate β values (Figure 1H). This is similar to low library complexity in low-input sequencing. The statistical uncertainty associated with this sampling process can be modeled with a binomial process conditioned on initial input, amplification cycles, and final signal intensity.

With the popularity of the human array, this technology has been rigorously tested by a very large number of independent labs. We took advantage of the extensive knowledge gleaned from the human array. In validating the accuracy of the DNA methylation measurements using titrations of synthetically methylated DNA, we observed a slight S-shaped deviation of the median methylation measurements compared with the identity line (Figure 2B). There was a similar slight offset in validation with WGBS of a biological sample (Figure 2C). This is likely attributable to the presence of background fluorescence intrinsic to the BeadChip assay platform, as observed in our human array studies,[20,99] resulting in slight measurement bias on the array platform. However, these minor inaccuracies compare favorably with the more polarized errors obtained with bisulfite sequencing without very deep coverage.

An advantage of using mice for epigenetic research is the ability to experimentally manipulate epigenetic control *in vivo*, which is not feasible in human subjects. Hypomorphic and full knockout alleles of the major maintenance DNA methyltransferase, *Dnmt1*, have been in use in the field for decades to determine the role of DNA methylation in tumorigenesis,[42,47] embryonic development,[38] X-inactivation,[100,101] transposon silencing,[102] and genomic imprinting,[103] among others. We have now for the first time documented the wider impacts of these allelic combinations across the methylome in different cell types. Homozygous knockout ESCs show substantial hypomethylation (median methylation reduced to 20%–55% compared with the wild-type average median, Figure 2D), but do retain some residual methylation, known to be attributable to the activities of the *de novo* methyltransferases *Dnmt3a* and *Dnmt3b*, which are expressed in mouse ESCs.[49,104,105] It is worth noting that the methylation levels are consistent with descriptions of clonal differences in homozygous hypomorphic N/N ESCs, with clone 10 displaying more severe

(D) Transcription factors whose binding sites are enriched in CpGs that gain methylation associated with tumorigenesis. Each dot corresponds to a transcription factor.

(E) Ontology terms enriched in target genes regulated by enhancers that gain DNA methylation.

(F) Transcription factors whose binding sites are enriched in CpGs that lose methylation associated with tumorigenesis.

(G) Scatterplot comparing age predicted by a mouse epigenetic clock using 347 CpG probes (y axis) and reported age (x axis) in disease-free tissues.

(H) Scatterplot comparing age predicted by a mouse epigenetic clock using 347 CpG probes (y axis) and reported age (x axis) in mouse intestinal tumors.

See also Figure S7 and Table S7.

hypomethylation than clone 52, consistent with reported differences in measured DNA methyltransferase activity published three decades ago.[38] Although severe *Dnmt1* depletion results in activation of Xist expression from the active X chromosome,[101] in the milder hypomorphic N/R combination, X-linked CpGs in female mice appear relatively resistant to genomic hypomethylation (Figure 3B), possibly reflecting selective pressure *in vivo* for the maintenance of a single active X chromosome.

As expected, imprinted loci display broadly intermediate levels of DNA methylation, consistent with their mono-allelic methylation status (Figure 3D). We observed tissue-specific losses of intermediate methylation among imprinted regions (Figure 3D) and differential susceptibility among imprinted loci to loss of methylation in *Dnmt1* hypomorphic conditions in ESCs (Figure S4D), suggesting variations among loci in the level of control of genomic imprints. Most imprinting loci displayed a complete loss of DNA methylation in ESCs with homozygous or compound-heterozygous *Dnmt1* mutations (Figure S4D). *Peg13* retained some residual methylation in *Dnmt1^N/N^* clone 52 but not in clone 10 (Figure S4D). It is interesting to note that all imprinted loci displayed considerably greater sensitivity to *Dnmt1* hypomorphic conditions in ESCs (Figure S4D) than in fetal and adult somatic tissues (Figure 3D). This may be a reflection of the strong dependence of imprint maintenance upon *Dnmt1* in early embryogenesis.[106,107] Testicular tissues appear to have the highest global methylation levels and are most resistant to loss of methylation in *Dnmt1* hypomorphic conditions (Figures 2E and S3A). This may be associated with *Dnmt3a* and *Dnmt3b* expression in the male germ-cell lineage.[108,109]

The prevalence of CpH methylation in adult brain tissues confirms prior reports.[61] The preference for CpA and, to a lesser extent, CpT, is consistent with this methylation mediated by Dnmt3A.[110] As part of this mouse DNA methylation atlas we have provided tissue-specific methylation signatures, which can be used in deconvolution of mixed tissue samples. The large number of samples incorporating a diversity of common confounding variables will also facilitate the use of reference-free deconvolution approaches.[111] We found that tissue-specific DNA hypomethylation was associated with binding of transcription factors involved in lineage specification and differentiation. For example, tail-specific hypomethylation is associated with binding of *Tbxt/Brachyury*, a transcription factor whose mutation leads to reduced tail length.[112] Brain-specific hypomethylation is enriched in binding sites of neuron development transcription factors such as *Neurog2*, *Foxg1*, and *Tbr2*. The powerful cell lineage tracing and reporter systems available for mice should facilitate further rapid advance of higher-resolution cell-specific signatures.

One of the concerns for many mouse-based disease models is how well molecular patterns represent those in human counterparts. We show that this Infinium mouse MM285 array can be used for analysis of hamster (11,505 probes) and rat (13,962 probes) samples, which provides access to other model organisms with some physiology more similar to that in humans. We also took care to include probes that target syntenic regions between human and mouse, to facilitate comparative epigenomics and to allow direct comparison of tumor-associated methylation changes in human and mouse lesions. These same probes are

also present on the human arrays, and a rich body of data on virtually all human tissue types for exactly the same CpG sites is readily available in the public domain, providing a comprehensive human reference. We found that species- and tissue-specific methylations were intertwined. The diversity of tissue types in our dataset allowed us to partially disentangle these effects. Residual tissue heterogeneity may confound our comparative analysis, as cell-type composition of corresponding organs is likely to vary somewhat among mammals. We anticipate that future studies using cell-type composition deconvolution, flow-sorted cells, and/or single-cell methylation profiling will be needed to confirm these suggestive findings. We also designed innovative methods to help dissect human and mouse signals from mixture samples such as PDX samples. These features serve to expand the utility of the platform and the bioinformatic analytical suite for modern mouse-based biomedical research.

We incorporated a selection of polymorphic genotyping probes with sets unique to each of 25 mouse strains, which will facilitate quality checks on strain purity and identity, benefit experiments with outbred mice, and assist with the identification of sample swaps, and which can be used to trace and accelerate backcrossing experiments. The diversity of tissues in our dataset allowed us to investigate the complex interactions between strain, tissue, age, and sex.

It has long been known that cancer-associated CpG-island hypermethylation in human tumors is enriched for regions occupied by Polycomb repressors in stem cells.[113–115] It was hypothesized that these events arise in an age-dependent manner and then accelerate in tumorigenesis.[116] Mouse models allow us to more tightly control genetic background and other confounding variables. We found that in *Apc^Min/+^* intestinal polyps, Polycomb-associated methylation is almost entirely an age-driven phenomenon, whereas tumor-associated methylation changes are associated with hypermethylation of enhancers associated with intestinal differentiation. This tumor model has a strong germline genetic driver, which may result in less reliance on epigenetic events in comparison with sporadic human cancers. Nevertheless, this mouse intestinal tumor model did recapitulate our earlier finding of loss of methylation in human colorectal cancer associated with binding sites for the AP-1 transcription factor.[71] This underscores both the relevance of this mouse model and the application of this DNA methylation platform.

We demonstrated the suitability of the mouse MM285 array to establish an epigenetic clock to estimate biological age. Since it is based on a wide diversity of tissue types, it should be relatively impervious to the tissue-specific effects encountered with clocks calibrated using a narrower selection of tissues. The fixed probe set and the high reproducibility of the methylation measurements provide a convenient and robust avenue to apply the clock to future datasets produced with this array.

The human Infinium arrays have been adapted to the analysis of hydroxymethylcytosine,[117–119] and we anticipate that this should be feasible for this array as well. In addition to the study of DNA methylation itself, we can infer tissue composition, biological age, genetic background, and sex from the same array data, further expanding the utility of the Infinium Mouse Methylation MM285 BeadChip.[18,20] These can be used as covariates of interest in analyses or to account for extra variance, and in turn

increase power and minimize confounding. In particular, tissue composition shift has long been raised as a common confounder in epigenetic studies.[120] Many of these covariates can also be used for examination of possible sample swaps, which can often occur in large-scale genomic studies. We anticipate that this mouse MM285 array will rapidly contribute large amounts of epigenetic data by taking advantage of the powerful mouse genetics[31,32] and complementing the extensive phenotype characterization.[92]

### Limitations of the study

The use of Infinium array technology has inherent limitations. Only a fixed set of CpGs with designed probes can be assayed and the assay does not provide allele-specific methylation measurements, nor is it compatible with ultra-low DNA input quantities, such as single-cell analysis, in its current implementation.[121] This study covered a wide swath of analytic approaches to genomic DNA methylation. As such, we were limited in the depth of analysis for each topic. This study is intended as an introduction to the features and applications of the platform and to provide a dataset resource so that other studies can delve more deeply into each topic area.

### STAR★METHODS

Detailed methods are provided in the online version of this paper and include the following:

- KEY RESOURCES TABLE
- RESOURCE AVAILABILITY
  - Lead contact
  - Materials availability
  - Data and code availability
- EXPERIMENTAL MODEL AND SUBJECT DETAILS
  - Experimental mice
- METHOD DETAILS
  - Mouse tissue dissection and processing
  - DNA bisulfite conversion, and array hybridization
  - DNA methyltransferase inhibition
- QUANTIFICATION AND STATISTICAL ANALYSIS
  - Probe designability
  - CpG probe selection – Genomic features
  - CpG probe selection – Target biology
  - CpG probe selection – Random selection
  - CpH and SNPs
  - Infinium BeadChip data preprocessing
  - Sex-specific methylation differences on X Chromosomes
  - Imprinting-associated and mono-allelic methylation
  - CTCF binding site methylation
  - Tissue-specific hyper- and hypomethylation signatures
  - Mapping the mouse array probes to other species
  - Human-mouse comparison
  - Estimation of the relative contributions of human and mouse DNA in patient-derived tumor xenografts (PDXs)

- Analysis of age- and tumor-associated DNA methylation
- Putative *Cdx2*, *Hnf4a*, and *Hnf4g* enhancer target genes affected by tumor-associated DNA hypermethylation
- Construction of epigenetic clocks

### SUPPLEMENTAL INFORMATION

### ACKNOWLEDGMENTS

This work was supported by NIH/NCI grants R01CA157918, R01CA212374, and R01CA234125 awarded to P.W.L. and by W.Z.'s startup fund at Children's Hospital of Philadelphia. H.S. is supported by NIH/NCI grant R37CA230748. P.A.J. is supported by NIH/NCI grant R35CA209859. This work was supported by the Vivarium and Transgenics Core, the Flow Cytometry Core, the Genomics Core, and the Scientific Computing Resources and Data Management team of Van Andel Institute. We thank Dustin Schones and Kevin Costello for support in defining VMRs.

### AUTHOR CONTRIBUTIONS

Conceptualization, W.Z., T.H., B.B., R.P., B.H.C., H.S., and P.W.L.; data curation, W.Z., T.H., B.B., K.K.F., J.A.P., P.E.S., M.S.B., H.S., and P.W.L.; formal analysis, W.Z., T.H., B.B., W.I., S.M.L., E.J.M., W.D., H.S., and P.W.L.; funding acquisition, W.Z., B.B., R.P., B.H.C., H.S., and P.W.L.; investigation, O.M., K.K.F., A.V., J.M.K., L.K., A.K.W., P.A.J., C.M.K., and M.A.; methodology, W.Z., T.H., B.B., P.A.J., C.M.K., M.A., H.S., and P.W.L.; project administration, W.Z., B.B., K.K.F., C.M.K., M.A., R.P., B.H.C., H.S., and P.W.L.; resources, K.-H.L., M.K., N.J.S., B.E., and N.A.V.S.; software, W.Z., T.H., B.B., W.I., S.M.L., E.J.M., W.D., and H.S.; visualization, W.Z., T.H., W.I., S.M.L., H.S., and P.W.L.; writing – original draft, W.Z., T.H., H.S., and P.W.L.; writing - review & editing, W.Z., T.H., M.K., J.A.P., M.S.B., A.K.W., P.A.J., C.M.K., R.P., B.H.C., H.S., and P.W.L.; supervision, W.Z., H.S., and P.W.L.

### DECLARATION OF INTERESTS

P.W.L. serves on the Scientific Advisory Board of AnchorDX. P.W.L. and H.S. serve on the Scientific Advisory Board of FOXO Technologies Inc. W.Z. receives research funding from FOXO Bioscience. B.B. and R.P. are employees and shareholders of Illumina, Inc. P.A.J. is a paid consultant for Zymo Research Corporation, whose bisulfite conversion kits were used in this research.

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

## STAR★METHODS

### KEY RESOURCES TABLE

| REAGENT or RESOURCE | SOURCE | IDENTIFIER |
|---|---|---|
| **Biological samples** | | |
| Patient-derived xenografts (PDXs) | Van Andel Research Institute | Human Colorectal Cancer PDX models:<br>VARI PDX-026<br>VARI PDX-034<br>VARI PDX-047<br>VARI PDX-048<br>VARI PDX-050<br>VARI PDX-065<br>VARI PDX-066<br>VARI PDX-067<br>VARI PDX-069<br>VARI PDX-075<br>VARI PDX-079<br>VARI PDX-081<br>VARI PDX-084<br>VARI PDX-104<br>VARI PDX-106<br>VARI PDX-085<br>VARI PDX-144 |
| Mouse DNA methylation calibration standards | EpigenDx | Cat# 80-8060M-PreMix |
| Rat DNA methylation calibration standards | EpigenDx | Cat# 80-8060R-PreMix |
| Human blood DNA | Bloodbank | Anonymous Donor |
| **Chemicals, peptides, and recombinant proteins** | | |
| 2'-deoxy-5-azacytidine (Decitabine, DAC) | Sigma-Aldrich | A3656 |
| **Critical commercial assays** | | |
| DNeasy Blood & Tissue Kit | Qiagen | Cat# 69506 |
| AllPrep DNA/RNA Mini Kit | Qiagen | Cat# 80204 |
| Mouse Methylation BeadChip | Illumina | Cat# 20041558 |
| Infinium HD FFPE DNA Restore Kit | Illumina | Cat# WG-321-1002 |
| EZ DNA Methylation Kit | Zymo Research | Cat# D5001 |
| **Deposited data** | | |
| Raw and analyzed data | This paper | GEO: GSE184410 |
| **Experimental models: Cell lines** | | |
| Hamster: CHO cells | ATCC | CRL-11268 |
| Mouse: NIH/3T3 cells | ATCC | CRL-1658 |
| Mouse: C3H 10T1/2 cells | ATCC | Clone 8, CCL-226 |
| Mouse: B16 melanoma cells | ATCC | CRL-6322 |
| Mouse: J1 ES cells and *Dnmt1* gene-targeted derivatives | Jaenisch Laboratory, Whitehead Institute for Biomedical Research | J1, N/N Cl52, N/N Cl10, S/S<br>Li et al., 1992[38] |
| Mouse: *Dnmt1* gene-targeted J1 ES Cells | Jaenisch Laboratory, Whitehead Institute for Biomedical Research | J1 C/C<br>Lei et al., 1996[49] |
| Mouse: *Dnmt1* gene-targeted J1 ES Cells | Laird Laboratory, University of Southern California | J1 R/+<br>Eads et al., 2002[47] |
| Mouse: *Dnmt1* gene-targeted J1 ES Cells | Laird Laboratory, University of Southern California | J1 PH/+, P/H<br>Chan et al., 2001[48] |

*(Continued on next page)*

*Continued*

| REAGENT or RESOURCE | SOURCE | IDENTIFIER |
|---|---|---|
| **Experimental models: Organisms/strains** | | |
| Mouse: 25 strains | The Jackson Laboratory | JAX # 002448 129S1/SvlmJ<br>JAX # 000646 A/J<br>JAX # 000648 AKR/J<br>JAX # 000651 Balb/Cj<br>JAX # 002282 BTBR_T+ ITpr3tf/J<br>JAX # 000659 C3H/HeJ<br>JAX # 002020 C57BL/6J<br>JAX # 000665 C57BL/10J<br>JAX # 005304 C57BL/6NJ<br>JAX # 000668 C57L/J<br>JAX # 000669 C58/J<br>JAX # 000928 CAST/EiJ<br>JAX # 000656 CBA/J<br>JAX # 000670 DBA/1J<br>JAX # 000671 DBA/2J<br>JAX # 001800 FVB/NJ<br>JAX # 002106 KK/HiJ<br>JAX # 000676 LP/J<br>JAX # 000550 MOLF/EiJ<br>JAX # 001976 NOD/ShiLtJ<br>JAX # 000684 NZB/BINJ<br>JAX # 002105 NZO/HILtJ<br>JAX # 001058 NZW/LacJ<br>JAX # 003715 PWK/PhJ<br>JAX # 001145 WSB/EiJ |
| Rat: CD® (Sprague Dawley) IGS Rat | Charles River Laboratories | Strain Code 001 |
| **Software and algorithms** | | |
| SeSAMe | Zhou et al., 2018[20] | https://bioconductor.org/packages/release/bioc/html/sesame.html |
| LOLA | Sheffield et al., 2016[75] | https://bioconductor.org/packages/release/bioc/html/LOLA.html |
| BEDTools | Quinlan et al., 2010[178] | https://github.com/arq5x/bedtools2 |
| **Other** | | |
| Mouse Methylation BeadChip Manifest and Probe Annotation | This paper | http://zwdzwd.github.io/InfiniumAnnotation |
| Comparison of the Illumina and SeSAMe probe Manifests | This paper | https://github.com/zhou-lab/InfiniumAnnotationV1/blob/main/Anno/MM285/MM285.manifest.comparison.tsv.gz |
| Mouse Genome ChromHMM Segmentation | Van der Velde et al., 2021 | https://www.encodeproject.org/ |
| ENCODE mouse WGBS and human MethylationEPIC data | ENCODE Project Consortium et al., 2020 | https://www.encodeproject.org/ |
| Ensembl Mouse Gene Model version 99 | Zerbino et al., 2018 | https://useast.ensembl.org/Mus_musculus/Info/Index |
| Transcription factor binding sites and histone modification data from the Cistrome Database | Liu et al., 2011 | http://cistrome.org/Cistrome/Cistrome_Project.html |
| Mouse strain genetic variation from The Mouse Genome Project | Keane et al., 2011 | https://www.sanger.ac.uk/data/mouse-genomes-project/ |
| Curation of the mouse array probes from imprinting control regions | This paper | https://github.com/zhou-lab/ImprintingAnno |
| 176 WGBS datasets | ENCODE, GEO | 176 Accession numbers listed in Table S1 |
| H3K27me3 ChIP-seq peaks | ENCODE | ENCODE accession numbers listed in Table S7 |

*(Continued on next page)*

*Continued*

| REAGENT or RESOURCE | SOURCE | IDENTIFIER |
|---|---|---|
| Probes used to estimate human/mouse contribution in PDX data. | This paper | Table S6 |
| Genes predicted to escape X-inactivation based on Mouse Methylation BeadChip | This paper | Table S2 |
| Inbred mouse strain genealogy | Beck et al., 2000 | https://www.nature.com/articles/ng0100_23 |

## RESOURCE AVAILABILITY

### Lead contact
Further information and requests for resources and reagents should be directed to and will be fulfilled by the lead contact, Peter W. Laird (peter.laird@vai.org).

### Materials availability
This study did not generate new unique reagents.

### Data and code availability
Infinium Mouse Methylation MM285 BeadChip manifest and annotations (including, gene association, chromatin state association, imprinting association, SNP-strain association etc.) are available at http://zwdzwd.github.io/InfiniumAnnotation#mouse. The epigenetic clock and data preprocessing pipelines are available through our R/Bioconductor software SeSAMe (https://bioconductor.org/packages/release/bioc/html/sesame.html). All DNA methylation data has been uploaded to Gene Expression Omnibus under the accession GSE184410.

## EXPERIMENTAL MODEL AND SUBJECT DETAILS

### Experimental mice
Mice were housed in an SPF facility with an institute-wide animal care program monitored by an Institutional Animal Care and Use Committee (IACUC) that adheres to the NIH Guide for Care and Use of Laboratory Animals, and fully accredited by the Council on Accreditation of the Association for Assessment and Accreditation of Laboratory Animal Care (AAALAC). We used both male and female mice in this study, and describe the influence of sex on molecular results, where appropriate. Mice were housed in Thoren™ individually ventilated caging systems on corncob bedding material. All cages were supplied with shredded paper as enrichment. Water was supplied through an RO/DI automated watering system. Mice were fed LabDiet® 5021 or 5010 *ad libitum*. Cages were changed once every 14 days by Vivarium husbandry staff.

## METHOD DETAILS

### Mouse tissue dissection and processing
Mice were euthanized and autopsied at various timepoints to allow the investigation of age-associated methylation differences. Tissue samples were dissected and snap frozen in liquid nitrogen. Mouse small intestines were processed either as full thickness samples or by stripping the mucosa. The distinction is annotated in the metadata. Full thickness samples were obtained by resecting the intestine proximally at the stomach and distally at the cecum. The sample was then incised longitudinally, rinsed, and divided into 10 sections of equal length, and then snap frozen in liquid nitrogen. Stripped mucosal tissue samples from the small intestine were processed by incubating the intestinal tissue segments at 37°C for 20 min in 30mM EDTA in HBSS. Following the incubation, samples were agitated on a vortex for 20 min. Following this, the residual muscularis and basement membrane was removed using forceps. The sample was then centrifuged at 400g for 5 min. The supernatant was removed and the remaining sample was snap frozen in liquid nitrogen. Genomic DNA was extracted from tissue samples using the DNeasy Blood & Tissue Kit (QIAGEN® # 69506) and resuspended in 10mM Tris buffer, pH 8.0. A similar procedure was also used to extract genomic DNA from PDX tumor samples stored in the Van Andel Institute biorepository. PBL were sorted using Beckman Coulter MoFlo Astrios sorter using the following markers: CD8 T cell (CD8$^+$ CD3$^+$), CD4 T cell (CD4$^+$ CD3$^+$), B cells (CD19$^+$, B220+) and monocytes (CD11b+, SSClo, Ly6G-).

### DNA bisulfite conversion, and array hybridization
DNA samples were quantified by Qubit fluorimetry (Life Technologies) and 500 ng each sample was prepared at a concentration of 11.1ng/uL in 45uL of volume for downstream bisulfite conversion. DNA samples were bisulfite converted using the Zymo EZ DNA Methylation Kit (Zymo Research, Irvine, CA USA) following the manufacturer's protocol with the specified modifications for the

Illumina Infinium Methylation Assay. After conversion, the bisulfite-converted DNA was purified using the Zymo-Spin binding columns and eluted in Tris buffer. Following elution, bisulfite-converted DNA was processed through the Infinium array protocol. Formalin-Fixed, Paraffin-Embedded (FFPE) tissues were prepared by fixation in 10% formalin for 24 or 48 h, as noted, followed by DNA extraction and bisulfite conversion. The bisulfite-converted DNA from FFPE samples was first processed using the Infinium HD FFPE DNA Restore kit workflow. To perform the Infinium assay, converted DNA was denatured with NaOH, amplified, and hybridized to the Infinium bead chip. An extension reaction was performed using fluorophore-labeled nucleotides per the manufacturer's protocol. Array BeadChips were scanned on the Illumina iScan system to produce IDAT files.

### DNA methyltransferase inhibition

C3H/10T1/2 cells (CCL-226) were seeded between the 9th and 15th passage at 2,000 cells/plate in 60 mm TC-treated dishes and dosed for 24 h with decitabine (200 nM, Sigma-Aldrich, A3656) or mock-treated with solvent (PBS). Cells were observed daily (24 days) by microscope for muscle formation. Cells were harvested on days 12 and 24 following start of drug treatment for DNA extraction with the DNeasy Blood and Tissue kit (Qiagen 69504).

## QUANTIFICATION AND STATISTICAL ANALYSIS

### Probe designability

Successful measurement of CpG methylation requires each Infinium probe to be uniquely hybridized to the mouse genome with high biochemical efficiency. Both hybridization and base extension should not be influenced by neighboring SNP or cytosine methylation. To ensure that all the probe selection conform to these prerequisites, we first developed an inclusion list of all designable CpGs, referred to as the designability list. For each CpG in the mouse genome, we investigated the two alternative Infinium-I probe design from the converted strand and map the probe sequence to the mouse genome using BISCUIT alignment software. To ensure uniqueness, we mapped the 30nt, 35nt, 40nt and the entire probe sequence and consider a probe to be uniquely mappable to the mouse genome if all subsequences align to the mouse genome with mapping quality greater than 20 for both the methylated (M-allele) and unmethylated allele (U-allele). 14,809,409 CpGs survived with at least one of the two probe designs passing this test. From this set, we further filter designs requiring no SNP and no additional CpG (besides the interrogated CpG) within 10nt from the probe's 3'-end. For SNP overlap, we retrieved the whole-genome variant calling of 15 in-bred mouse strains (129P2/OlaHsd, 129S1/SvImJ, 129S5, A/J, AKR/J, BALB/cJ, C3H/HeJ, C57BL/6NJ, CBA/J, DBA2/J, FVB/NJ, LP/J, NOD/ShiLtJ, NZO/HlLtJ, WSB/EiJ) from the Mouse Genome Project.[122] We excluded SPRET/EiJ, PWK/PhJ and CAST/EiJ from the overlapping analysis due to their divergence from the laboratory mice, and thus too high SNP density. We further required fewer than six CpGs in the entire probe sequence, to avoid suboptimal hybridization caused by the variable methylation status at neighboring CpGs. We calculated a 0-1-ranged design score for each Infinium design to encapsulate its hybridization efficiency as determined by GC content and melting temperature. We only retain designs over 0.3 in design score. Applying all the above filtering leads to 9768540 CpGs with at least one viable probe design.

### CpG probe selection – Genomic features
#### Promoters

We downloaded 141,767 transcripts for *Mus musculus* from the Ensembl database release 96.[123] For each transcript, we first collapsed transcripts into 54,376 distinct transcription start sites (TSSs) for protein coding transcripts. For each TSS, we define promoter CpGs as the CpGs within 1,500 bp from both direction of the TSS. We ranked these sites based on their distance to the TSS and only considered CpG sites in the designable list. When ranked with higher quality sites (score $\geq 0.6$), lower quality sites (score <0.6) are given an extra penalty of 500, so that a lower-quality site is only preferred when it is closer to all the high-quality site by over 500bp with respect to their distances to the TSS. From the top of the ranking, we chose at most two CpG sites for each TSS. After pooling with probes from other design categories, we ultimately included 100,948 CpGs for 54,376 TSSs. This selection leaves 1,972 (3.6%) TSSs not associated with any CpG in the array. We also included CpGs for transcripts of other "bio-types", particularly lincRNA, pseudogene, and miRNA, as defined in Ensembl. We identified 8,621 unique lincRNA TSS of which 1,327 do not have an associated CpG in the designability set. We chose at most two CpGs per TSS minimizing the distance to the TSS. A similar selection was performed for pseudogene and miRNA transcripts, leading to a total of 15,030 lincRNA CpGs, 10,339 pseudogene CpGs and 4,222 miRNA CpGs. See Figure S1A for a summary of these categories in the final manifest.

#### Enhancers

We took a multi-step approach to select enhancer probes. We first collected ChromHMM segmentation of the mouse genome from two prior studies[124,125] of mouse embryonic stem cells and eight mouse tissues. We identified all the high-quality probes that overlap with enhancer segments (both strong and poised enhancers). We then removed probes located within 1 kbp from the TSS of a protein-coding transcript. Finally, we studied methylation levels of these CpG sites in a compendium of 176 publicly available mouse DNA methylomes (Table S1). To focus on tissue-specific enhancers, we selected CpG sites observed to be heavily methylated (>0.7 in methylation level) in over five samples while unmethylated (<0.3 in methylation level) in five samples. We were able to narrow down to a set of 757,439 CpGs that display such sample-specificity. We randomly chose a subset of 60,000 CpGs resulting in 58,759 CpG probes after infiltering and quality control to represent tissue-specific enhancer elements. We also included 1,247 CpGs covering enhancers validated in the VISTA database.[126]

🔗 CellPress

**Cell Genomics**
**Resource**

### CTCF binding sites

CTCF binding scaffolds the 3D chromatin conformation.[127] DNA methylation at CTCF binding sites is known to be inversely associated with CTCF binding.[128] We downloaded the CTCF ChIP-seq data from the mouse ENCODE project.[129] For evidence of CTCF binding sites we require a site to meet the following criteria. First the CpG is located inside in CTCF ChIP-seq peaks from at least two in 19 tissue types.[129] The binding peaks were validated with chromHMM segmentation reported by the same previous study.[125] We further created a ladder of evenly sampled subgroups of CpGs by their methylation status in a compendium of 176 mouse WGBS datasets (Table S1). We randomly sampled 1,000 CpGs from each bin from the 10 bins of DNA methylation fraction in the 0-1 interval. We excluded sites that always exhibited 90-100% methylation in all the samples. This selection strategy yielded 8,616 probes designed to capture the CTCF binding status.

### CpG islands

To capture CpG islands, particularly those that are not associated with annotated transcript promoters, we selected one designable CpG for each CpG island associated with gene promoter and three for each CpG island that are not overlapping with any of the transcription start sites annotated in the mouse Ensembl database (release 96). CpG islands annotations were downloaded from UCSC genome browser database (mm10). For CpG islands that overlap with transcription start sites, the selection would enhance the representation of these CpG-island-associated TSSs. In total, 17,134 CpGs were included to cover 13,365 of the 16,023 CpG islands in the mouse genome.

### Gene bodies

Gene body DNA methylation is known to be associated with gene expression.[50,54] To capture gene body methylation, we focused on canonical protein-coding transcripts defined in RefSeq.[130] Gene body regions were defined as genomic intervals spanning from 2 kb downstream of the TSSs, excluding 5' CpG islands (as defined by UCSC genome browser) to the transcription termination site. Transcripts that are shorter than 2 kb were excluded. We randomly selected one designable CpG from each transcript. This approach selects 25,011 CpG sites to target gene bodies. After merging with other design categories, we reached a total of 111,702 non-promoter genic probes in the final manifest. of these, 83,731 CpGs fall into intronic regions, while 403 CpGs are in close proximity to a splice site (+/- 2 bp).

### Transposable elements

Transposable elements (TEs) make up 40% of the mouse genome. TE DNA methylation states are often used to represent genome-wide DNA methylation levels[131] and for tracking developmental stage and tissue of origin.[132,133] We downloaded the transposable element consensus sequence from RepBase.[134] We only focused on major rodent-specific repeat families with more than 1000 copies in the mouse genome. We then selected probes to target internal CpGs of these transposable elements using the consensus sequence. Further restriction by probe design scores reduces to a final inclusion of 4,723 repetitive element probes. Of note, 14 probes in this selection target mouse B1 elements, 242 probes target 18 mouse IAP families, and 423 probes target 100 mouse L1 transposable element families (including both full-length copies and fragments).

### Sex chromosomes

To enable methylation analysis of the sex chromosome DNA, we expanded the representation of chromosome X and Y with a random sampling of 15,174 chromosome X from the designability list. Due to the suboptimal quality and the high repetitive nature of the Y chromosome genome sequence, Y chromosome CpGs are under-represented on the array. We included all designable chromosome Y CpGs to compensate this underrepresentation leading to a total of 3,780 Y-chromosome CpG probes.

### Mitochondrial CpGs

We included 32 mitochondrial CpGs. Since mitochondrial DNA is unmethylated due to lack of access to DNA methyltransferases, we expect that these CpGs can be used for technical control.

### CpG probe selection – Target biology

### Imprinted biology and mono-allelic methylation

Unlike most genomic CpGs, a small fraction of CpGs tend to be consistently mono-allelically methylated in diverse tissue types, yielding a beta value of approximately 0.5. To target imprinted DMR, we collected 26 known imprinted regions (Key resources table) and included 661 high-quality CpGs located in these regions. We used the same WGBS sample compendium (Key resources table, excluding germ cell samples and early embryonic samples) to identify consistently intermediately methylated sites. Chromosome X CpGs were excluded and analyzed separately (see above). We call CpGs as consistently intermediately methylated if the CpG is measured in more than 6 samples and over 65% of the samples shows methylation level between 0.35 to 0.65. Overlapping these CpGs with high-quality CpGs yields 7,813 CpGs that passed our design criterion. Since the placental tissue is known to have genomic imprinting in a distinct set of loci, we also selected 981 probes that displayed intermediate methylation level one placenta tissue sample.[135]

### Germ cell and early embryonic development

Global epigenetic remodeling occurs in germ cell and early embryonic development in mammalian species to reset the epigenome during intergenerational transition.[136] This is epitomized by two waves of global loss of methylation in pre-implantation embryo and primordial germ cells, except for a small subset of CpGs targeting endogenous viral elements.[137] In addition, each stage of the remodeling could be characterized by a unique DNA methylation signature.[133,138] To capture this biology, we selected CpGs observed to be either specifically methylated or unmethylated in germ cells, including primordial germ cells,[139,140] oocytes[141] and spermatocytes, zygotes, placenta,[142] and multiple early embryo stage samples.[143]

### Epigenetic clock, aging and cancer

DNA methylation dynamics is known to inform organismal and cellular aging and is implicated in age-associated diseases such as cancer. It has been used to predict chronological age and biological age in multiple tissues.[72] To capture age-associated DNA methylation change in this array, we analyzed five existing datasets that reported six mouse arrays[75–79] using other methylation assay technologies. We included all the 765 CpG sites that had been identified as designable CpG sites from our above analysis. To capture DNA methylation aberration in cancer, we compared a previously published DNA methylome dataset of two colon cancer samples induced by Dextran Sulfite Sodium (DSS) and Azoxymethane,[144] as well as two colon normal samples.[135,144] We identified CpG sites that are methylated (methylation level >0.7) in the two cancer samples but not in the two normal colon samples (methylation level <0.3). Only CpGs with sequencing depth greater than 6 reads in all the samples were considered. This analysis leads to 8,330 CpG sites included in the design for association with colon cancer-specific methylation changes. Our previous studies suggested that CpGs that are relatively isolated and are flanked by C/G (the so-called solo-WCGW CpGs) in the late replicated genomic territory, also known as partially methylated domains (PMDs), are more likely to lose methylation during mitotic cell division.[145–147] Solo-WCGW CpGs are defined by CpGs flanked by a C or G and without additional CpGs in the flanking sequences 35bp in length, a distance shown to be most predictive of methylation alteration during cellular aging. The annotation of common PMDs was obtained from our previous study.[145] To further confirm that these regions are late in DNA replication, we used nuclear lamina-associated domains determined in a previous study[148] as surrogates. We included 5,095 randomly chosen solo-WCGW PMD CpGs.

### Metastable epi-alleles, variably methylated regions

Variation in DNA methylation at endogenous retroviruses such as IAP can cause inter-individual and potentially trans-generational epigenetic inheritance.[57–59] These variably methylated CpG sites have the potential to function as a sensor of the environment and have been implicated in obesity susceptibility.[149] We collected 6,402 VM-IAP regions from a prior study[56] and included one CpG site for each of the VM-IAP, yielding 5,849 probes targeting this category. 633 regions were left out due to lack of designable CpGs.

### Human-mouse synteny

We started by projecting human probes already included in the HumanMethylation EPIC arrays[150] to mouse using the UCSC LiftOver utility.[151,152] 29,054 of the EPIC CpGs map to designable CpGs in the mouse genome. Of these, 28,719 syntenic CpGs map uniquely in both the human and mouse genome. Note that the human CpG and the syntenic mouse CpG are not necessarily queried using the same probe design. These CpGs may have different flanking sequence in the two species and hence different probe sequences. We labeled each probe as well as their associated syntenic human EPIC array probe. Because the human EPIC array enriches for gene promoters which are generally more conserved than rest of the genome, we also observe an enrichment of promoter probes for this category.

## CpG probe selection – Random selection

We added 28,011 randomly selected designable CpG sites with the intent of covering uncharacterized biology and sex chromosomes which are otherwise under-represented.

## CpH and SNPs

### Non-CpG cytosines (CpHs)

Non-CpG cytosine methylation has been found to be implicated in modulating gene expression in human tissues.[153] We included 2,310 CpH sites evenly distributed by sequence context to the CpA, CpC and CpT groups. Half of these CpH sites were chosen for being methylated in the mouse brain tissue based on a previous WGBS study.[61] The other half of the CpHs were randomly chosen. Like CpG probes, all the CpH probes were quality controlled by requiring a unique mapping of the probe sequence to the bisulfite-converted mouse genome and having a high design quality score.

### Strain-specific SNPs

The inclusion of strain-specific probes will allow the investigator to assess or verify the genetic background and lineage composition of inbred mice. We studied currently available whole-genome sequence data of 36 inbred and wild mouse strains from the Mouse Genome Project.[122] We curated the lineage history (Figure 6A) of these strains by adapting the known lineage history.[154] We selected at most ten representative SNPs from each branch of this phylogeny, with some SNPs queried using multiple design variants. We excluded repetitive elements (as marked in RepeatMasker)[155] and sites with no additional SNP within 25 base pairs and no CpG within 30 base pairs. This leads to a list of 1,485 SNP probes included in the designed mouse methylation array to target 591 branch-specific SNPs.

### The basic parameters of the mouse methylation array

The mouse array is approximately half the size of the HM450 array (Figure S1B). It covers 1.3% of the mouse genomic CpGs, a ratio slightly lower than the HM450 array. The Basic Parameters of the Mouse Methylation Array: 95.8% of the probes on the mouse array are CpG probes. The rest are CpH, SNP, control probes and probes that did not pass quality control. Compared to the two previous generations of the human arrays, the mouse array included fewer CpGs and slightly fewer CpH probes (Figure S1C). It includes more SNP probes and control probes. 4,541 probes that do not meet design objective (sequence mismatches and suboptimal hybridization performance, representing probes with uk in the probe name prefix) were also exposed to the users. In our default SeSAMe processing pipeline, these probes were masked with NA. The masking can be optionally removed. The total number of probes sums up to 296,070.

We have previously shown that the out-of-band fluorescence color channel can capture background signal better than the internal negative control probes, constituting an effective data normalization strategy for background subtraction[99] and detection p value calibration.[20] Probes designed with Infinium-I chemistry provide 61,873 out-of-band signals for these applications (Figure S1D).

The mouse array also contains 2,874 control probes, including negative control probes and bisulfite conversion control probes (Figure S1E).

The mouse array contains a small fraction of probes targeting the replicated daughter strand, as opposed to the parent bisulfite converted strand, which is an innovation compared to the human Infinium arrays (Figure S1F). This provides additional design flexibility to cover challenging genomic regions.

Most probes can be mapped uniquely to the mouse genome (Figure S1G). A small fraction of 7,364 CpG probes were intentionally designed to target multiple regions. The default setting for our SeSAMe processing pipeline[20] masks 4,541 probes that do not meet design objectives (sequence mismatches and suboptimal hybridization performance), but this masking can be optionally removed (Figure S1G right panel).

We have mapped the mouse methylation probes to both the mm10 and mm39 genome builds, as represented in a confusion matrix in Figure S1H.

Unlike the human array in which all probes were designed to target a single CpG, the mouse array provided design redundancy for a small fraction of CpGs and SNPs (Figure S1I). This allows us to investigate variation between different designs. To reflect the probe design redundancy, we expanded the probe ID system with a four-letter suffix to accommodate information needed to uniquely identify a probe (Figure S2A). The suffix captures information of whether the read is designed against top or bottom strand of the 122-mer template DNA, whether the design is targeting converted or the synthesized (opposite) strand, whether the probe is of the Infinium-I or Infinium-II chemistry and finally an integer enumerating different replicate of the same probe.

### Infinium BeadChip data preprocessing

Mouse array IDATs were preprocessed using SeSAMe.[20] We first calculated detection p values for each probe using the pOOBAH algorithm.[20] We then performed background subtraction using the noob method,[99] followed by a dye bias correction using the dyeBiasCorrTypeINorm function provided in the SeSAMe package. Signal intensities were then summarized into beta values using the getBetas function, with multimapping probes and probes with insignificant detection p value masked (p > 0.2). We investigated probes with detection p value between 0.01 and 0.2 using cross-validation and found them largely reflecting true biological methylation signal.

### Sex-specific methylation differences on X Chromosomes

To identify the methylation difference between active and inactive X chromosome in female cells, we studied the methylation difference between male and female samples. Since male samples have one active X while female cells have both an active and an inactive X chromosome, variation in female DNA methylation compared to male tissues reflects the difference of inactive X chromosome methylation compared to active X. We performed a linear regression of DNA methylation on tissue type, sex and strain and calculated the slope coefficients of sex-specific effect of the methylation of X-linked CpGs, which is equivalent to the sex-specific methylation difference depicted in Figure 3C. Most CpG island CpGs display hypermethylation in the females while CpGs located in inactive-X-specific long-non-coding RNAs are hypomethylated in females (Figure 3C). We identified 6 genes covered by the MM285 array predicted to escape X-inactivation[55] (Table S2).

### Imprinting-associated and mono-allelic methylation

Using criteria of different degrees of stringency, we identified four groups CpGs potentially involved in mono-allelic methylation. Group I includes CpGs frequently (>50%) found to be intermediately methylated (beta >0.3 and beta <0.7) across 138 somatic tissue samples. Group II further requires the CpGs to be intermediately methylated in over 90% of the samples and of full (>0.7) or no methylation (<0.3) in the three testis samples which are largely composed of spermatocytes. Group III and Group IV requires the CpGs to be located at annotated imprinting control regions (ICRs) or secondary DMRs (sDMRs), or just ICRs only, respectively. The imprinting control regions were retrieved from previous studies.[156–170]

### CTCF binding site methylation

CTCF binding sites were retrieved from 11 primary tissue data from ENCODE (Table S3). CTCF motif scanning was done within the peak sequence using FIMO[171] and JASPAR motif MA0139.1.[172] Methylation signals are then aligned with respect to the start position of the CTCF motifs. We further filtered CTCF binding sites with more than 0.2 in methylation level in all the tissues within 100bp window flanking the CTCF motif. This results in 5,449 CTCF binding sites left for analysis (Table S4).

### Tissue-specific hyper- and hypomethylation signatures

We used a one-vs-rest approach to identify CpGs uniquely hypo- or hypermethylated in each tissue. We first computed the area under the curve (AUC) for discriminating the target tissue from tissues of other tissue types. We considered probes capable of perfectly discriminating the target tissue from the rest. We then ordered the probes by comparing the target tissue and the other tissues in DNA methylation. The top 200 probes in the methylation difference were regarded as tissue methylation signatures. The signatures were

then inspected for enrichment with transcription factor binding. Each signature CpG set was studied for its overlap with transcription factor binding sites retrieved from the Cistrome database.[66] The statistical significance of the overlap was evaluated using Fisher's exact test. To investigate the interaction of multiple DNA methylation-determining factors, we performed multiple multivariate regression[173] of DNA methylation on four predictors, i.e., tissue, strain, sex and age, using 467 non-tumor samples using the DML function in the SeSAMe package.[20] For each predictor, we performed an F-test and considered probes with a p-value smaller than 0.05 and effect size (delta beta value) greater than 0.1 as differentially methylated CpGs.

### Mapping the mouse array probes to other species
We collected 310 Ensembl (version 101)[123] whole genome sequences. Each sequence was indexed for bisulfite read alignment using BISCUIT tolerating one mismatch. For Infinium-I probes, allele A and allele B were mapped separately to the genome. The mapping positions were evaluated for consistency between the two alleles. Extension base was checked for targeting CpG methylation measurement. Phylogenetic trees were retrieved from NCBI taxonomy. We filtered and plotted 56 species in Figure 5A including 12 muroida, 11 other rodentia, 8 primates, 9 other mammals, 13 other vertebrates and 3 invertebrates.

### Human-mouse comparison
To investigate human-mouse epigenetic conservation, we collected 33 datasets with 8 tissues in humans, matching the 41 tissue datasets in mouse from ENCODE and other previous studies (Table S5). We merged DNA methylation measurements on the 28,719 syntenic CpGs with unique mappings in both species interrogated using EPIC and the mouse array independently. We performed a linear regression analysis of the DNA methylation measurements on both species and tissue types, allowing for statistical interaction of the two factors. To define tissue- and species-specific methylation, we first calculated the mean methylation for each combination of tissue and species using multivariate regression. The tissue-specific methylation effect was defined as the maximum methylation mean subtracted by minimum methylation mean across different tissue levels (regardless of species). Likewise, the species-specific methylation effect was defined as the maximum methylation mean subtracted by the minimum methylation mean between the two species (regardless of tissue). Figure S6E plots the tissue- (X axis) and species-methylation effect (Y axis) of the 28,719 syntenic CpGs. CpGs with tissue-specific effect >0.4 and tissue/specific >3 are colored blue. Likewise, CpGs with species-specific effect >0.4 and species/tissue-specific effect >3 are colored red. The two may match in the cases of statistical interaction. High density of data points on the diagonal line in Figure S6E suggests extensive interaction of tissue- and species-specific effect.

### Estimation of the relative contributions of human and mouse DNA in patient-derived tumor xenografts (PDXs)
We developed two different methods to quantify the relative contributions of human and mouse in a mixed DNA sample. The first method uses 19 human-mouse syntenic Infinium-I probes with single-nucleotide variations at the extension bases between human and mouse (Table S6). These probes emit signals in one fluorescent color channel when hybridized to human DNA and the other when hybridized to mouse DNA. We calculated the ratio of the signal from the color channel corresponding to the human allele to the sum of the signals from both color channels [Human/(Human + Mouse)] for each probe. We then took the median of the ratios from the 19 probes. In the second method, we ran the same DNA samples on the mouse and human (EPIC) arrays and analyzed 259,626 and 733,164 non-syntenic probes in the mouse and human arrays, respectively. We calculated the ratio of the median total signal intensities from the human array over the sum of median total signal intensities from the human and mouse arrays [Human/(Human + Mouse)] in a sample. To create a standard curve, we extracted mouse DNA from fat and spleen tissues and mixed them with different proportions (0, 0.05, 0.1, 0.15, 0.25, 0.5, 0.75, 0.85, 0.9, 0.95 and 1) of human blood DNA. We fitted LOESS curves between the signal ratios [Human/(Human + Mouse)] and the known proportions of human DNA (Figures S6F and S6G). We generated the standard curve for each method based on the mean of the two LOESS fitted values from the fat and spleen DNA (Figures S6H and S6I) and used them to estimate human DNA fraction in 17 colorectal cancer PDXs (Figure 5G).

### Analysis of age- and tumor-associated DNA methylation
We analyzed DNA methylation profiles in small intestinal tumors and tumor-adjacent small intestinal mucosa excised from $Apc^{Min/+}$ mice. We examined DNA methylation levels at the Mir200c promoter, which is methylated in mesenchymal cells but unmethylated in epithelial cells, to select 56 tumors and 13 normal tissues consisting primarily of epithelial cells (beta value < 0.2) (See Figure 4E). We excluded probes designed for X and Y chromosomes and having missing data in more than 25% of the samples. We further filtered out probes constitutively unmethylated (maximum beta value < 0.2) or constitutively methylated (minimum beta value > 0.8) across all samples. The remaining 92,738 probes were used in further analysis. Multivariate linear regression was used at the probe level to identify CpG sites that showed DNA methylation changes associated with age or tumor after adjusting for mouse strains. The p values were corrected for multiple comparisons using the Benjamini-Hochberg method. Probes showing tumor-associated methylation changes were selected based on the adjusted p value <0.01 and absolute coefficient value for tumor >0.1, with positive and negative coefficients indicating hypermethylation (n = 4,429) and hypomethylation (n = 3,398), respectively. Similarly, age-associated methylation changes were defined as having adjusted p value <0.01 with the signs of the coefficient for age specifying either hypermethylation (n = 3,585) or hypomethylation (n = 990). Probes located in the PcG target sites (Figure 7A) were determined using H3K27me3 ChIP-seq replicated peaks on postnatal 0-day mouse intestine downloaded from the Gene Expression Omnibus (GEO) under

accession GSE82655.[67] To examine transcription factors that bind to the CpG sites subjected to age- or tumor-associated DNA methylation changes, we analyzed 9,066 ChIP-seq data on 704 individual DNA binding factors curated in the Cistrome Data Browser (DB).[174] The statistical significance of enrichment for transcription factor binding sites among different groups of CpGs was determined using Fisher exact test with 200bp regions centered on the target CpGs using the R package *LOLA*.[175] p-values were adjusted for multiple comparisons using the Benjamini-Hochberg method.

### Putative *Cdx2*, *Hnf4a*, and *Hnf4g* enhancer target genes affected by tumor-associated DNA hypermethylation

We identified 1,295 significantly hypermethylated CpGs overlapping the binding sites for at least one of the intestine-specific transcription factors, *Cdx2*, *Hnf4a*, and *Hnf4g*. To gain a better insight into the impact of tumor-associated DNA hypermethylation at these sites, we investigated putative target genes of these transcription factors. We collected the regulatory potential (RP) scores calculated in Cistrome DB, which were assigned to each RefSeq gene and reflect its likelihood of being regulated by a particular factor.[174] We considered at most ten nearest genes within 1,000kb upstream and ten nearest genes within 1,000kb downstream from the CpG sites. We then investigated the list of genes with RP scores above 2 in at least one ChIP-Seq data for all three transcription factors. GO terms over-representation analysis was performed using the *enrichGO* function with default parameters as implemented in the R package *clusterProfiler*.[176]

### Construction of epigenetic clocks

The elastic-net regularized linear model was built using glmnet.[177] To select most predictive CpGs, we set alpha to 0.5 and lambda to 0.098, selected using the cv.glmnet function which automatically optimizes the mean absolute error of the model using 10-fold cross validation. This procedure leads to an epigenetic clock composed of 347 CpG probes with an estimated mean absolute error of 1.19 months. For testing the enrichment of polycomb-targeted CpGs, we downloaded 77H3K27me3 ChIP-seq narrow peaks from ENCODE (Table S7) and identified genomic CpGs that overlaps with each peak. We used Fisher exact test to evaluate the statistical significance of the overlap between the 347 clock CpGs and genomic CpGs associated with H3K27me3. The annotation for the 347 probes, along with their weights, have been incorporated into the current version of SeSAMe.

