## [Document S2. Transparent peer review records for Zhou et al. · Cell Genomics]

Title: DNA Methylation Dynamics and Dysregulation
Delineated by High-Throughput Profiling
in the Mouse

Author list: Wanding Zhou^{1,2*}, Toshinori Hinoue³, Bret Barnes⁴, Owen Mitchell³, Waleed Iqbal¹, Sol Moe Lee¹, Kelly K. Foy³, Kwang-Ho Lee³, Ethan J. Moyer¹, Alexandra VanderArk⁵, Julie M. Koeman⁶, Wubin Ding¹, Manpreet Kalkat³, Nathan J. Spix³, Bryn Eagleson⁷, John Andrew Pospisilik³, Piroska E. Szabó³, Marisa S. Bartolomei⁸, Nicole A. Vander Schaaf^{3,5}, Liang Kang³, Ashley K. Wiseman³, Peter A. Jones³, Connie M. Krawczyk⁵, Marie Adams⁶, Rishi Porecha⁴, Brian H. Chen⁹, Hui Shen^{3*}, Peter W. Laird^{3,10}

Summary

Initial submission: Received : November 8th, 2021

Scientific editor: Sonia Mulyil

First round of review: Number of reviewers: 3
Revision invited : February 23rd, 2022
Revision received : March 20th, 2022

Second round of review: Number of reviewers: 3
Accepted : May 20th, 2022

Data freely available: Yes

Code freely available: Yes

This transparent peer review record is not systematically proofread, type-set, or edited. Special characters, formatting, and equations may fail to render properly. Standard procedural text within the editor's letters has been deleted for the sake of brevity, but all official correspondence specific to the manuscript has been preserved.

Referees' reports, first round of review

Reviewer 1:

In this manuscript Zhou et al. developed a mouse Infinium DNA methylation array, which they call MM285, which is the mouse equivalent equivalent to the widely popular human 450K array. They started with 9,768,540 designable CpGs in the mouse genome after rigorous filtering, and then designed a total of 297,415 probes that target various genomic features to capture DNA methylation dynamics across diverse contexts. They showed that the data generated using this array are highly reproducible across different experimental and laboratory settings, and for a wide range of input DNA quantities including FFPE samples. They performed thorough technical and biological validations to confirm that the array can produce the expected results robustly. Finally, they used this array to profile 1,239 samples encompassing distinct tissues, strains, age, sex, and pathologies. They further demonstrated the utility of this rich resource focused on the following biological contexts:

1. Comparative epigenomics and its application to PDX assessment.
2. Backcross tracing
3. Identification of age-associated and tumor-associated methylation changes.
4. Construction of a mouse epigenetic clock.

Interesting findings include tissue-specific methylation correlating with lineage specific transcription factors binding, age-associated hypermethylation sites enriched at H3K27me3 regions, while age-associated hypomethylation sites enriched at sites by the cohesin complex. Overall, the manuscript is well written; the analysis is thorough and rigorous; and the results are presented clearly and effectively. It is an exceptional resource for the mouse genetic community in general, and for researchers studying epigenetic processes using mouse models. I recommend its publication after the authors address the following minor critiques:

1. "We explored the dominant factors influencing variations in the mouse methylome by projecting each of the 1,076 non-tumor samples to 2D space, using t-distributed stochastic neighbor embedding (t-SNE) applied to all methylation probes ..." - Here the authors applied t-

SNE to the methylation profiles and later claimed that the sample clustering was dominated by tissue type, followed by sex strain, and age. Since the authors did not explain how they reached to this conclusion, I assume it was based on visual inspection of the t-SNE embedding. Like UMAP, t-SNE was originally designed for data visualization purpose. Although these visuals are useful for hypothesis generation, one should avoid solely relying on it to draw conclusions as t-SNE does not preserve global structure well and may not faithfully represent the relationships among the data. Later in the text the author did mention they tried to perform multivariate regression to more quantitatively measure the effects of tissue, strain, sex and age on CpG methylation. However, this analysis was not shown in Figure 4B as it was referred to, nor other places in the manuscript.

2. "Pairwise correlation between human and mouse tissues in the syntenic CpG methylomes revealed a varying degree of tissue correspondence" - While this comparison between human and mouse is interesting, in-depth analysis or discussion is needed to better interpret the result. First, how many tissue-specific methylated regions are in synteny between human and mouse? It is difficult to interpret this result without knowing the degree to which the differences are driven by the exclusion of signature regions of those tissues from the correlation calculation. Second, would the tissue heterogeneity be a contributing factor here? Biologically, are the cell type compositions of tissues largely similar between human and mouse? Technically, the way the tissue was isolated and prepared will likely affect the cell type composition in the sample, which may be a confounding factor here. These questions can be addressed by future single-cell studies but should be discussed in this manuscript.

3. In the section named "Biological Validation of the DNA Methylation ...", the authors used "substantial", "appreciable", etc., at many places. It would be good to add in some quantification of these observations by performing statistical test.

4. In Figure 4A, what do the colors represent? Please add legends to the plot.

5. In Figure S5B, hierarchical clustering of 16 samples was shown. I'm wondering how and why these 16 samples were chosen? Also, a dendrogram should be added to this plot to help readers better understand the relationships between samples.

6. Figure 4B shows the intersections of differentially methylated CpGs. I could not find the text describing how the differential test was performed.

Data and Software availability

I was unable to get access to the data using the GEO accession number (GSE184410) provided by the authors. At the time of writing, the data is still private. The code has been deposited to Github under the MIT license. It is also available as a R package on Bioconductor.

Reviewer 2:

The authors present the design and initial application of a new mouse Infinium array (mm285) for higher throughput DNA methylation investigation. There is no question that the array will be welcome by users in the DNA methylation community and hence the paper is a useful

reference/resource for that purpose.

The study is overall well executed and covers most, if not all, questions one may have on what the array can do, so there is really not much to add what could be tested. I can see the rationale for going through all this at a rather shallow level. Every topic is covered a bit, but few new insights are presented. For each one could readily see further analysis that would possibly feature the strength of the array further, so I wonder whether at least one topic could be showcased further to lead to further insights. That said, it may not be necessary as the aim is mostly to introduce the resource to the field.

I would suggest moving a few more panels to the Supplement and improving the presentation of the main figures (fewer panels and more clarity).

Reviewer 3:

Zhou et al. present here the development and assessment of a mouse Infinium DNA methylation array containing 297,415 probes capturing mouse DNA methylation diversity. Moreover, the authors present a mouse DNA methylation atlas comprised by 1,239 DNA samples of different tissue-types and conditions that can be used as a reference tool. Furthermore, authors describe the application of the mouse methylation array for different purposes such as the study of genomic imprinting, comparative epigenomics or backcross tracing among others. Importantly, authors report methylation changes associated with aging, differentiation and tumorigenesis, highlighting the relevance and impact that the introduction of this novel mouse methylation array will have on the scientific community considering the importance of this model organism.

The manuscript makes significant contributions to the literature with its comprehensive analyses and introduction of a novel array to study the mouse DNA methylome. However, there are major concerns that must be addressed.

Major comments:

1.-Authors must provide access to raw data deposited in the Gene Expression Omnibus. Accession "GSE184410" is currently private. A reviewer token was not supplied and data is not accessible to evaluate the manuscript. It is a must.

2.-Numbers do not match. The number of probes for the different design categories of the mouse Infinium BeadChip in figure S1A do not match the numbers depicted in the Methods Details under the CpG Probe Selection - Genomic Features section. For instance, regarding promoter probes (page 22), authors state that "we were able to select 71,240 CpGs for 54,376 TSSs". However, 100,948 TSS probes appear on Figure S1A. Considering that only a small percentage of CpG probes are redundant (Figure S1I), this difference requires further explanation.

Most importantly, regarding gene bodies (page 23), authors state that "This approach selects 25,542 CpG sites to target gene bodies, leading to 150,981 non-promoter genic probes in the final manifest. 125,726 CpGs fall into intronic regions". I am confused about these numbers. How is possible that 25,542 CpG sites targeting gene bodies lead to more than 150,000 non-promoter genic probes including more than 125,000 CpGs in intronic regions? Furthermore, the definition of gene bodies varies between the manuscript "2kb downstream the TSSs" (page 23) and Figure S1A ("Gene body CpGs at least 1500bp downstream from TSSs"). Which definition have the authors considered to select these probes?

The discrepancy between numbers remains for the majority of design categories (i.e., page 22 "12,270 lincRNA CpGs, 10,493 pseudogenes CpGs and 4,058 miRNA CpGs" vs 15,030 lincRNATSS, 10,339 PseudogeneTSS and 4,222 miRNA probes depicted in Figure S1A).

3.-The proper annotation of the final manifest file is key for users to correctly analyze mouse Infinium BeadChip data. Thus, when compared to Illumina's Infinium Mouse Methylation Manifest File ("<https://emea.support.illumina.com/content/dam/illumina-support/documents/downloads/productfiles/mouse-methylation/infinium-mouse-methylation-manifest-file-csv.zip>") important discrepancies are shown. Illumina's manifest file contains 287,692 probes compared to the 297,415 probes described in the manuscript. For instance, important differences are found in the number of control probes between both Illumina's and manuscript's manifest files regarding the number of control probes (642 vs 4,216 respectively). May the authors be able to explain these discrepancies? Considering that the MM285 mouse array is already accessible to the research community, the reasons behind these important differences must be elucidated for a proper and correct use of the array.

4.-In the analysis for CpG sites associated with colon cancer-specific methylation changes, authors used a previously published DNA methylome dataset of a colon cancer sample (Abu-Remaileh et al., 2015). Considering that only three replicates of one single cancer colon sample are included in this dataset, I was wondering whether this sample size is large enough to effectively discover differentially methylated CpG sites between the different conditions. In this regard, I was also wondering how the differential methylation analysis was performed to obtain the 8,488 CpG sites included in the array design.

5.-On page 5, authors state that "enhancers are the most overrepresented on the MM285 array". However, when looking at the pie chart at the bottom, quiescent chromatin regions seem to be the most overrepresented regarding the % of CpGs on the array. Please, may the author be able to further explain these results?

Minor comments:

1.-Regarding the human-mouse synteny, since exceptional conservation of synteny can reflect important functional relationships between genes, I was wondering whether authors may have

explored the possibility of having used a conservation score to further filter the number of CpGs in this category and to provide further information that may be useful for the scientific community when trying to establish functional assessments between both human and mouse DNA methylomes.

2.-Mouse array IDATs were preprocessed using SeSAMe and signal intensities were summarized into beta values with multimapping probes and probes with insignificant detection p-value masked ($p > 0.2$). Do the authors believe this p-value is stringent enough? I worry that this p-value threshold may be too loose, especially when looking at Figure 5C, where the probe success rate of probes designed to work only for human species remains impressively high up to almost a mixture of 100% mouse DNA. May the authors further explain how it is possible that with a 75% mouse DNA content, around 90% of probes succeeded when hybridized to the Human EPIC array? And how does this result correlate with authors statement on page 4 "only 1% of Infinium Methylation EPIC probes are mappable to the mouse genome?".

3.-When mapping the mouse array probes to other species (Figure 5A), what did the authors consider as functional probes? As detailed on the methods, alignment was performed using BISCUIT, did authors allow for any mismatches, and if so, for how many?

4.-Sometimes is difficult to find the reference of the samples used for particular DNA methylome analyses under the Key Resources Table. For instance, in page 22, authors claim that "176 publicly available mouse DNA methylomes (see Key Resources Table for a list of data sources)", but I was not able to find this particular reference for this publicly available dataset of 176 samples.

5.-Please provide more details on the software used to conduct the different multivariate linear regression analyses.

6.-Regarding the construction of epigenetic clocks, authors claim to have used 77 H3k27me3 ChIP-seq peaks from ENCODE detailed in Table S1. However, only 11 files are found on Table S1.

7.-May the authors be able to provide the list of 6 genes predicted to escape X-inactivation and the list of 19 human-mouse syntenic probes used to estimate relative contributions in PDXs? I believe this would be very useful for the scientific community.

8.-The "33 ENCODE" data sets used to investigate human-mouse epigenetic conservation include datasets from outside ENCODE project (Table S3).

Authors' response to the first round of review

Reviewer #1

I recommend its publication after the authors address the following minor critiques: 1. "We explored the dominant factors influencing variations in the mouse methylome by projecting each of the 1,076 non-tumor samples to 2D space, using t-distributed stochastic neighbor embedding (t-SNE) applied to all methylation probes ..." - Here the authors applied t-SNE to the methylation profiles and later claimed that the sample clustering was dominated by tissue type,

followed by sex strain, and age. Since the authors did not explain how they reached to this conclusion, I assume it was based on visual inspection of the tSNE embedding. Like UMAP, t-SNE was originally designed for data visualization purpose. Although these visuals are useful for hypothesis generation, one should avoid solely relying on it to draw conclusions as t-SNE does not preserve global structure well and may not faithfully represent the relationships among the data. Later in the text the author did mention they tried to perform multivariate regression to more quantitatively measure the effects of tissue, strain, sex and age on CpG methylation. However, this analysis was not shown in Figure 4B as it was referred to, nor other places in the manuscript. We concur that 2D tSNE projection does not preserve distances in the original high dimensional data space. To further quantify the contribution of different biological variables to sample clustering, we have performed density-based clustering of the original data using DBSCAN and calculated the Uncertainty Coefficient of the clustering result, given each variable. We have now added a new Supplemental Fig S5B, to illustrate this result (included here as well as Response Figure 1). The Uncertainty Coefficient quantifies the amount of information entropy in one variable explained by another variable. Consistent with our visual inspection, tissue type explains the most information in clustering. Age is the second most influential factor on clustering after tissue type, highlighting a global impact of aging on DNA methylation determination. We have added the following text to the revised text to discuss this result: Page 9, line 381: "We further clustered these DNA methylomes using DBSCAN and calculated the uncertainty coefficient using cluster membership and sample meta information. Consistent with the tSNE analysis, tissue type is the dominating factor of DNA methylome determination, followed by age, strain, and sex etc. (Figure 4A right, Figure S5A, S5B). Hierarchical clustering of 246 samples representing 22 primary tissue types revealed grouping primarily by tissue types (Figure S5C), verifying the importance of tissue type in methylome determination." We thank the reviewer for pointing out that we did not fully describe Figure 4B in the first submission. To clarify, we have added the following text to the manuscript: Response Figure 1 - Supplemental Figure S5B. Response to Reviewers 2 Page 9, line 387: "We then performed whole array multivariate regression analysis of CpG methylation on tissue, strain, sex and age in 467 non-tumor samples (Figure 4B). The number of CpGs showing significant DNA methylation differences associated with each variable and their interactions are shown in a Venn diagram. Consistent with our unsupervised analysis, tissue type and age were the strongest individual drivers of specific methylation behavior. Many CpGs displayed joint influences from multiple covariates on methylation levels. For example, methylation at 95% of the tissue-specific CpGs is under joint influence by other factors, including strain (Figure 4B)." 2. "Pairwise correlation between human and mouse tissues in the syntenic CpG methylomes revealed a varying degree of tissue correspondence" - While this comparison between human and mouse is interesting, in-depth analysis or discussion is needed to better interpret the result. First, how many tissue-specific methylated regions are in synteny between human and mouse? It is difficult to interpret this result without knowing the degree to which the differences are driven by the exclusion of signature regions of those tissues from the correlation calculation. Second, would the tissue heterogeneity be a contributing factor here? Biologically, are the cell type compositions of tissues largely similar between human and mouse? Technically, the way the tissue was isolated and prepared will likely affect the cell type composition in the sample, which may be a confounding factor here. These questions can be addressed by future single-cell

studies but should be discussed in this manuscript. We identified 217,601 probes with some evidence of tissue-dependent DNA methylation (absolute regression slope >0.2). Among these, only 18,492 (8.5%) are CpGs syntenic to human EPIC array CpGs (defined as CpGs in the EPIC arrays and mappable to valid mouse CpGs based on the UCSC liftOver mm10- to-hg38 chain file) (Hinrichs et al., 2006; Kuhn et al., 2013), and 36,607 (16.8%) were evolutionarily conserved (PhastCons > 0.8). We can only conduct a joint comparison of tissue- and species-specific methylation behavior on CpGs in synteny between human and mouse. Therefore, we can only conduct this analysis on a small fraction of probes available on the array. From among 28,719 human-mouse syntenic CpGs, 17,341 CpGs show evidence of tissue dependence (absolute regression slope >0.2). The fraction of CpGs with tissue-dependent methylation among syntenic CpGs ($17,341$ (slope >0.2) / $28,719$ = 60%) is high, but not as high as the fraction of CpGs with tissue-dependent methylation on the entire array ($217,601$ (slope >0.2) / $284,862$ = 76%). Therefore, tissue-dependent methylation behavior is slightly underrepresented among syntenic CpGs. There could be multiple explanations for this. One is that the syntenic CpGs were sourced exclusively from among probes already represented on the human EPIC array. Differences in design approaches between the EPIC content and our mouse content could be one reason for this difference. Another contributing factor could be that the most conserved syntenic regions may control fundamental processes that are universal across cell types, and therefore less cell-type specific. Whatever the underlying reasons, this analysis is indeed too complex to explore in detail in this first manuscript. However, we do agree with the reviewer that this question is interesting, even if the analysis of human-mouse similarities in tissue-specific methylation is complex. One way in which we addressed this complexity is by including interaction terms in our multivariate regression. Our multivariate model includes tissue as a covariate and models the statistical interaction between tissue and species explicitly (i.e., DNA methylation \sim species + tissue + tissue * species). In Response Figure 2 we plot the distribution of the tissue-related regression coefficient (including interaction). We also agree with the reviewer that a full comparison of evolutionary epigenetic conservation should resolve tissue heterogeneity and would ideally be conducted at the cell type or single-cell resolution. The relative similarity of cell type composition for each tissue in different species is an assumption that is likely imperfect for many tissues. However, organ function, physiology, histology, and anatomy are conserved to a certain degree among mammals. In our opinion, the interesting findings in this initial analysis are worth Response Figure 2 - Tissue Dependency of HumanMouse syntenic CpGs. 3 reporting, and serve to kindle the interest of other researchers for a more thorough investigation of this issue. We modified and expanded the text in the Discussion to clarify and better address this issue: Page 15, line 698: "We found that species- and tissue-specific methylation were intertwined. The diversity of tissue types in our dataset allowed us to partially disentangle these effects. Residual tissue heterogeneity may confound our comparative analysis, as cell-type composition of corresponding organs is likely to vary somewhat among mammals. We anticipate that future studies using cell-type composition deconvolution, flow-sorted cells, and/or single-cell methylation profiling will be needed to confirm these suggestive findings." 3. In the section named "Biological Validation of the DNA Methylation ...", the authors used "substantial", "appreciable", etc., at many places. It would be good to add in some quantification of these observations by performing statistical test. We intentionally avoid the term "significantly" in the manuscript where a statistical test is not

warranted, for example when we are comparing methylation levels among specific individual cell lines, and therefore intentionally use the types of descriptors mentioned by the reviewer. However, we have attempted to address the reviewer's concern by adding additional quantitative information for some of these comparisons. We modified the following text segments: Page 7, Line 253: "Heterozygous Dnmt1 knockout ES cells did not show a clear reduction in DNA methylation levels, consistent with the presence of a remaining wild-type allele in the heterozygotes, but the median methylation of various homozygous and compound heterozygous allelic combinations ranged from 20% to 55% of the average median of the wild-type lines (Figure 2D)." Page 7, Line 284: "We repeated this experiment with 5-aza-2'-deoxycytidine (decitabine, DAC) and observed substantial hypomethylation, with the median beta value of DAC-treated cells reaching 81% of the median beta value of mock-treated DNA at Day 12, with some recovery by day 24 to 86% of mock-treated cells (Figure 2F), consistent with prior reports (Taylor and Jones, 1979; Yang et al., 2014)." Page 15, Line 649: "Homozygous knockout ES cells show substantial hypomethylation (median methylation reduced to 20-55% compared to the wild-type average median, Figure 2D), but do retain some residual methylation, known to be attributable to the activities of the de novo methyltransferases Dnmt3a and Dnmt3b, which are expressed in mouse ES cells (Lei et al., 1996; Okano et al., 1999; Okano et al., 1998)." We also removed the use of "substantial" when accurate numbers are not given (Lines 71, 743). 4. In Figure 4A, what do the colors represent? Please add legends to the plot. We regret this oversight. The colors in Figure 4A represent tissue type and sex, with colors consistent with other figures. The color legend is the same as in Figure 3A. We added the following text in the caption of Figure 4A. Page 20, Line 817: "Figure 4 (A) t-SNE cluster map showing samples clustered by tissue type (left) and sex (right). The color legend is the same as in Figure 3A." 5. In Figure S5B, hierarchical clustering of 16 samples was shown. I'm wondering how and why these 16 samples were chosen? Also, a dendrogram should be added to this plot to help readers better understand the relationships between samples. 4 The 16 samples were chosen from the earliest batch of produced data for simplicity of visualization. However, we agree that a more thorough study is warranted. We have now expanded this analysis to 246 normal samples and replaced the former Figure S5B by a new Figure S5C (this new version is included here as Response Figure 3). The new figure includes dendrograms, as suggested. Unsupervised clustering of genome-wide methylation clusters primarily by tissue type. We have also updated the corresponding text in the discussion: Page 9, Line 385: "Hierarchical clustering of 246 samples representing 22 primary tissue types revealed grouping primarily by tissue types (Figure S5B)," Page 32, Line 1390: "Matrix representing hierarchical clustering of pairwise Spearman correlation coefficients of global methylomes of 246 samples representing 22 different tissue types." 6. Figure 4B shows the intersections of differentially methylated CpGs. I could not find the text describing how the differential test was performed. We thank the reviewer for pointing out this lack of clarity. We have added the following text to the Methods section for this analysis. Page 28, Line 1219: "To investigate the interaction of multiple DNA methylation-determining factors, we performed multiple multivariate regression (Robertson et al., 2021) of DNA methylation on four predictors, i.e., tissue, strain, sex and age, using 467 non-tumor samples using the DML function in the SeSAMe package (Zhou et al., 2018b). For each predictor, we performed an F-test and considered probes with a P-value smaller than 0.05 and effect size (delta beta value) greater than 0.1 as differentially methylated CpGs." Data and

Software availability I was unable to get access to the data using the GEO accession number (GSE184410) provided by the authors. At the time of writing, the data is still private. The code has been deposited to Github under the MIT license. It is also available as a R package on Bioconductor. We regret neglecting to include a reviewer's token in our first submission. Please use this token: Gjixsucutzwzhkz The full dataset will be made publicly available upon acceptance of the paper for publication.

Reviewer #2

The study is overall well executed and covers most, if not all, questions one may have on what the array can do, so there is really not much to add what could be tested. I can see the rationale for going through all this at a rather shallow level. Every topic is covered a bit, but few new insights are presented. For each one could readily see further analysis that would possibly feature the strength of the array further, so I wonder whether at least one topic could be showcased further to lead to further insights. That said, it may not be necessary as the aim is mostly to introduce the resource to the field. I would suggest moving a few more panels to the Supplement and improving the presentation of the main figures (fewer panels and more clarity). Response Figure 3 - Supplemental Figure S5C 5 We appreciate and share these sentiments expressed by the reviewer. We indeed struggled to balance showcasing the utility and potential of the array with a deeper dive into some of the applications. In the end, we concluded that the reader would be best served by being made aware of the vast number of specialized features, such as strain SNPs, imprinted loci, human/mouse comparisons, CpH probes, etc. We fear that moving panels advertising any of these types of applications to the supplement would result in their being overlooked by most readers. As this reviewer noted, in the end "...the aim is mostly to introduce the resource to the field." That being said, we respectfully suggest that our Figure 7 does do a deep dive into age- and tumor-associated methylation in more detail, and provides quite a few new insights and advances in the field.

Reviewer #3

Major comments: 1. Authors must provide access to raw data deposited in the Gene Expression Omnibus. Accession "GSE184410" is currently private. A reviewer token was not supplied and data is not accessible to evaluate the manuscript. It is a must. We regret neglecting to include a reviewer's token in our first submission. Please use this token: Gjixsucutzwzhkz The full dataset will be made publicly available upon acceptance of the paper for publication. 2. Numbers do not match. The number of probes for the different design categories of the mouse Infinium BeadChip in figure S1A do not match the numbers depicted in the Methods Details under the CpG Probe Selection - Genomic Features section. For instance, regarding promoter probes (page 22), authors state that "we were able to select 71,240 CpGs for 54,376 TSSs". However, 100,948 TSS probes appear on Figure S1A. Considering that only a small percentage of CpG probes are redundant (Figure S1I), this difference requires further explanation. The numbers that were listed in the original Methods section refer to categories of design intent, whereas Figure S1A is a recounting of categories in the final array. We failed to make this distinction

clear in the first submission. The numbers the reviewer mentioned from the original Methods section correspond to CpGs we selected for each design objective in the first round of CpG selection. After the initial selection, probes for different design categories were combined and further filtered in several rounds of post-processing (e.g., determination of strand, Infinium chemistry and replicates etc.) and experimental validation. The numbers that appear in Figure S1A represent a recounting of CpGs after pooling of all design categories. For example, 71,240 CpGs were included to cover TSS during the initial design. Other TSS-associated probes were included for other design objectives (which likely overlapped with TSS, e.g., EPIC-array conserved CpGs), leading to a total of 100,948 CpGs included at TSS vicinities. To avoid potential confusion for the reader, we fully revised our Figure S1A to include the recounted numbers only. We updated the corresponding numbers in the revised method section to reflect the same number we show in Figure S1A (Line 962 to line 1208). Some of the updates to the numbers also reflect the change from a recent Illumina manifest since our initial submission (version A1 to A2, as pointed out by the reviewer in the following comment). We added the following text to clarify the promoter probe pooling: Page 23, Line 972: "After pooling with probes from other design categories, we ultimately included 100,948 CpGs for 54,376 TSSs." Most importantly, regarding gene bodies (page 23), authors state that "This approach selects 25,542 CpG sites to target gene bodies, leading to 150,981 non-promoter genic probes in the final manifest. 125,726 CpGs fall into intronic regions". I am confused about these numbers. How is possible that 25,542 CpG sites targeting gene bodies lead to more than 150,000 non-promoter genic probes including more than 125,000 CpGs in intronic regions? Furthermore, the definition of gene bodies varies between the manuscript "2kb downstream the TSSs" (page 23) and Figure S1A ("Gene body CpGs at least 1500bp downstream from TSSs"). Which definition have the authors considered to select these probes? The discrepancy in numbers is again attributable to the pooling and overlap in design categories mentioned above. The 25,542 CpGs were selected to target gene bodies during the initial category design (we used at least 2kb downstream of the TSS). After pooling of all design categories and recounting gene body probes, we ended up with quite a few more probes. To be consistent with the design, we recalculated the number of gene body CpGs using a consistent 2 kb definition. We have updated the Methods section on Gene Bodies to read as follows: Page 24, Line 1,013: "Gene Bodies | Gene body DNA methylation is known to be associated with gene expression (Baubec et al., 2015; Yang et al., 2014). To capture gene body methylation, we focused on canonical protein-coding transcripts defined in RefSeq (Tatusova et al., 2016). Gene body regions were defined as genomic intervals spanning from 2 kb downstream of the TSSs, excluding 5' CpG islands (as defined by UCSC genome browser) to the transcription termination site. Transcripts that are shorter than 2 kb were excluded. We randomly selected one designable CpG from each transcript. This approach selects 25,011 CpG sites to target gene bodies. After merging with other design categories, we reached a total of 111,702 non-promoter genic probes in the final manifest. of these, 83,731 CpGs fall into intronic regions, while 403 CpGs are in close proximity to a splice site (+/- 2 bp)." The discrepancy between numbers remains for the majority of design categories (i.e., page 22 "12,270 lincRNA CpGs, 10,493 pseudogenes CpGs and 4,058 miRNA CpGs" vs 15,030 lincRNATSS, 10,339 PseudogeneTSS and 4,222 miRNA probes depicted in Figure S1A). We have updated Figure S1A and the following Method section with recounted numbers to avoid confusion. Page 23, Line 977: "A similar selection was performed for

pseudogene and miRNA transcripts, leading to a total of 15,030 lincRNA CpGs, 10,339 pseudogene CpGs and 4,222 miRNA CpGs. See Figure S1A for a summary of these categories in the final manifest.” 3. The proper annotation of the final manifest file is key for users to correctly analyze mouse Infinium BeadChip data. Thus, when compared to Illumina’s Infinium Mouse Methylation Manifest File (["https://emea.support.illumina.com/content/dam/illumina-support/documents/downloads/productfiles/mouse-methylation/infinium-mouse-methylation-manifest-filecsv.zip"](https://emea.support.illumina.com/content/dam/illumina-support/documents/downloads/productfiles/mouse-methylation/infinium-mouse-methylation-manifest-filecsv.zip)) important discrepancies are shown. Illumina’s manifest file contains 287,692 probes compared to the 297,415 probes described in the manuscript. For instance, important differences are found in the number of control probes between both Illumina’s and manuscript’s manifest files regarding the number of control probes (642 vs 4,216 respectively). May the authors be able to explain these discrepancies? Considering that the MM285 mouse array is already accessible to the research community, the reasons behind these important differences must be elucidated for a proper and correct use of the array. Although we designed the content of this array, we do not control the marketing or technical information for the array released by Illumina. We outline below several categories of probes with design flaws currently in the Illumina A2 manifest that should be omitted, and useful probes that are missing that should be included. We have notified Illumina of these issues, and Illumina is working on a new update of the manifest. In the meantime, we maintain our own independent SeSAmE manifest for this reason, which is available at: <https://github.com/zhoulab/InfiniumAnnotationV1/tree/main/MM285> In comparing our manifest to the most up-to-date Illumina manifest (version A2), we identified 9,858 probes unique to our manifest (referred to as the “SeSAmE-unique” set) and 838 probes unique to Illumina’s 7 manifest (referred to as the “Illumina A2-unique” set). The SeSAmE-unique set includes 2,310 CpH probes, 2,874 control probes, 133 rs probes (SNP probes) and 4,541 uk-probes (unknown, including design flaws) that have been omitted from the Illumina A2 manifest. The discrepant probe IDs can be found at <https://zhouseriver.research.chop.edu/InfiniumAnnotation/current/MM285/MM285.manifest.comparison.tsv.gz> We discuss here the reasons for our inclusions and exclusions. The 2,310 CpH probes were missed by Illumina, but we evaluated their performance, and found signal strength and low methylation status in our test samples to match our expectations from the biology (Figure below, left panel). The 133 rs-probes were excluded from the Illumina A2 manifest because they are missing official rs-IDs, but are instead indexed by the genomic location. We have included these in the SeSAmE manifest. We flagged the 4,541 unknown probes not labeled as such in the Illumina A2 manifest as containing design flaws. Since some of these probes can be useful for normalization purposes, we chose to reveal these to the community user, but to flag them as “unknown”. Since they are not labeled as CpG, CpH, or SNP probes, they will automatically be excluded from most biological analyses, but they are retrievable to skilled users who would like to use them for advanced purposes. Upon investigating the chip addresses of these probes, we found that 838 of these mis-designed probes are retained in the Illumina A2 manifest and are designated as CpH probes (rather than uk probes). We found that their beta value distribution revealed a high methylation readout, which is unlikely for CpH cytosines, consistent with our interpretation of design flaws (see Response Figure 4, right panel). The use of the Illumina manifest would lead to incorrect biological conclusions for these

mis-designed flawed CpH probes. The details of the control probes and their use are proprietary to Illumina, but the SeSAmE-unique control probes are accessible in IDAT files. Hence, we decided to expose these in SeSAmE for potential community use (e.g., by normalization methods such as Funnorm). In summary, all CpG probes are identical between the two manifests. The discrepancy largely originates from our manifest identifying more probes with design flaws, as well as making more probes accessible for analysis (design flaws and control probes). We host a file on github and on our annotation website to illustrate this discrepancy: <https://github.com/zhou-lab/InfiniumAnnotationV1/tree/main/MM285> <https://zhouseriver.research.chop.edu/InfiniumAnnotation/current/MM285/MM285.manifest.comparison.tsv.gz> 4. In the analysis for CpG sites associated with colon cancer-specific methylation changes, authors used a previously published DNA methylome dataset of a colon cancer sample (Abu-Remaileh et al., 2015). Considering that only three replicates of one single cancer colon sample are included in this dataset, I was wondering whether this sample size is large enough to effectively discover differentially methylated CpG sites between the different conditions. In this regard, I was also wondering how the differential methylation analysis was performed to obtain the 8,488 CpG sites included in the array design. We agree with the reviewer that the small sample size has limited our ability to comprehensively cover cancer-associated differentially methylated regions (e.g., acknowledged in Line 616 of the revised manuscript). We were limited by the available mouse whole-genome bisulfite sequence datasets available at the time of the array design. In our analysis, we compared two colon adenoma samples (both from Abu-Remaileh et al. 2015) and two normal colon samples (including one normal sample from Abu-Remaileh et al. 2015 and one from Hon et al. 2013). We looked for CpGs with sequencing coverage greater than 6, showing high DNA Response Figure 4. CpH Probes Inappropriately Included or Excluded in the Illumina A2 Manifest. 0.0 0.1 0.2 0.3 0.4 0.5 0 5 10 15 20 25 30 2310 ch probes Ilmn Mft misses N = 2310 Bandwidth = 0.002929 Density 0.2 0.4 0.6 0.8 1.0 0 2 4 6 8 838 ch probes in Ilmn Mft that should be uk N = 838 Bandwidth = 0.01501 Density 8 methylation in both tumor samples (>0.7) and low DNA methylation in both normal samples (< 0.7) in the two cancer samples but not in the two normal colon samples (methylation level < 0.3). Only CpGs with sequencing depth greater than 6 reads in all the samples were considered. This analysis leads to 8,330 CpG sites included in the design for association with colon cancer-specific methylation changes." Page 14, Line 615: "We also randomly selected CpGs to enable discovery of novel biology in the event of limited mouse WGBS data, e.g., colon cancer-associated datasets." 5. On page 5, authors state that "enhancers are the most overrepresented on the MM285 array". However, when looking at the pie chart at the bottom, quiescent chromatin regions seem to be the most overrepresented regarding the % of CpGs on the array. Please, may the author be able to further explain these results? Although the quiescent probes are the largest category by absolute number, they are underrepresented on the array compared to that category of CpGs in the entire genome. When we normalize the number of MM285 probes by the total number of genomic CpGs in each chromatin state, enhancers are the most overrepresented and quiescent chromatin CpGs are relatively underrepresented on the array. We clarified this distinction in the revised manuscript. The relevant text now reads: Page 5, Line 176: "We derived consensus chromatin states from 66 ENCODE chromHMM calls (van der Velde et al., 2021) and plotted the abundance of probes on the array representing each chromatin category (Figure 1C, pie chart). Although quiescent

chromatin probes represent a large fraction of the probes, normalization to the number of CpGs for each chromatin state in the entire genome reveals that enhancers and promoters are the most overrepresented on the array, compared to their relative abundance in the entire genome, underscoring the utility of the array in characterizing gene transcription control (Figure 1C, lollipop plot, Figure S2B)."

Minor comments:

1. Regarding the human-mouse synteny, since exceptional conservation of synteny can reflect important functional relationships between genes, I was wondering whether authors may have explored the possibility of having used a conservation score to further filter the number of CpGs in this category and to provide further information that may be useful for the scientific community when trying to establish functional assessments between both human and mouse DNA methylomes. We thank the reviewer for this suggestion. In the revision, we provided 60-way PhastCon and PhyloP annotation for the MM285 probes, and made them available through our annotation site <http://zwdzwd.github.io/InfiniumAnnotation>. Response Figure 5 (Fig S6B in the revised manuscript) illustrates the sequence conservation between syntenic CpGs and CpGs from other design categories. Syntenic CpGs (designed to map to EPIC CpGs) are the 2nd most conserved design category in PhastCons score, next only to mitochondrial probes, confirming the synteny and the conservation score. We included the following discussion of this analysis: Page 11, Line 467: "As expected, CpGs designed to map to EPIC CpGs are more evolutionarily conserved than the other nuclear DNA CpG categories (Figure S6B). 2. Mouse array IDATs were preprocessed using SeSAmE and signal intensities were summarized into beta values with multimapping probes and probes with insignificant detection p-value masked ($p > 0.2$). Do the authors believe this p-value is stringent enough? I worry that this p-value threshold may be too loose, especially when looking at Figure 5C, where the probe success rate of probes designed to work only for human species remains impressively high up to almost a mixture of 100% mouse DNA. May the authors further explain how it is possible that with a 75% mouse DNA content, around 90% of probes succeeded when hybridized to the Human EPIC array? And how does this result correlate with authors statement on page 4 "only 1% of Infinium Methylation EPIC probes are mappable to the mouse genome?". We thank the reviewer for this keen observation. When a DNA sample consisting of 75% mouse tissue is run on the HumanMethylationEPIC array, the probe success rate is indeed close to 90% (Figure 5C). The Infinium platform can tolerate a high fraction of non-hybridizing DNA from other species. The 25% of the DNA sample that is human-derived is likely sufficient to yield a 90% probe success rate. However, to investigate whether an insufficiently stringent p-value is perhaps contributing to the high probe success rate, we replotted the figure adjusting the detection p-value threshold to a stringent 0.01 (Response Figure 6, showing data for the MM285 run on the left and EPIC run on the right). Despite observing a slight drop in the overall probe success rate in mouse probes for mouse tissues or EPIC probes for human tissues (indicating a too stringent detection p value threshold), we still observed that 74% of the human-only probes reported detection success in the 75% mouse tissue samples. This suggests that

the detection p-value threshold is not the primary cause of the high probe success rate in samples with a high fraction of DNA from a different species. Response Figure 5 - Supplemental Figure S6B. Response Figure 6. 10 We further investigated the beta value distribution of the probes that fall in the range between 0.01 and 0.2 in detection p-value. We selected all probes with a [0.01,0.2] detection p-value in a 90% mouse-10% human sample (high-p-value set) and all probes with

Referees' report, second round of review

Reviewer 1:

The authors have addressed my concerns to the previous version. The revised manuscript is suitable for publication now.

Reviewer 2:

No further comments. The manuscript presents a valuable resource.

Reviewer 3:

The manuscript was thoroughly revised. It addressed most of the concerns that I had previously raised. Specially, I would like to recognize author's effort to help better understand discrepancies with Illumina's manifest file and to further investigate whether a p-value > 0.2 threshold was stringent enough to effectively detect probe success rate.

However, there are two key points that must be addressed before I can recommend its publication.

Major comments:

- 1.- Raw data is still not accessible. The reviewer's token provided (Gjixsucutzwhkz) to access GSE184410 dataset is not valid. Authors must provide a valid reviewer's token to access raw data.
- 2.- Authors may include in the manuscript and/or in the KEY RESOURCES TABLE a reference to the link:

<https://zhouseriver.research.chop.edu/InfiniumAnnotation/current/MM285/MM285.manifest.comparison.tsv.gz>

Otherwise, Illumina's manifest discrepant probe IDs file won't be available and accessible to researchers.

Minor comments:

- 1.- Authors may update numbers in Figure S1C to match the total number of 296070 described in Figure S1A.
-

Authors' response to the second round of review

We have addressed the remaining concerns of Reviewer 3 as follows:

1. We have made the raw data public. It is now fully and freely accessible at the following URL :

<https://www.ncbi.nlm.nih.gov/geo/query/acc.cgi?acc=GSE184410>

2. We have added the manifest comparison URL to the Key Resources Table, as requested.